# Statistical Efficiency of Distributional Temporal Difference Learning

Yang Peng[*]           Liangyu Zhang[†]           Zhihua Zhang[‡]

## Abstract

Distributional reinforcement learning (DRL) has achieved empirical success in various domains. One core task in the field of DRL is distributional policy evaluation, which involves estimating the return distribution $\eta^\pi$ for a given policy $\pi$. The distributional temporal difference learning has been accordingly proposed, which is an extension of the temporal difference learning (TD) in the classic RL area. In the tabular case, Rowland et al. [2018] and Rowland et al. [2024a] proved the asymptotic convergence of two instances of distributional TD, namely categorical temporal difference learning (CTD) and quantile temporal difference learning (QTD), respectively. In this paper, we go a step further and analyze the finite-sample performance of distributional TD. To facilitate theoretical analysis, we propose non-parametric distributional temporal difference learning (NTD). For a $\gamma$-discounted infinite-horizon tabular Markov decision process, we show that for NTD we need $\widetilde{O}\left(\frac{1}{\varepsilon^{2p}(1-\gamma)^{2p+1}}\right)$ iterations to achieve an $\varepsilon$-optimal estimator with high probability, when the estimation error is measured by the $p$-Wasserstein distance. This sample complexity bound is minimax optimal up to logarithmic factors in the case of the 1-Wasserstein distance. To achieve this, we establish a novel Freedman's inequality in Hilbert spaces, which would be of independent interest. In addition, we revisit CTD, showing that the same non-asymptotic convergence bounds hold for CTD in the case of the $p$-Wasserstein distance for $p \geq 1$.

## 1 Introduction

In high-stake applications of reinforcement learning (RL), such as healthcare [Lavori and Dawson, 2004, Böck et al., 2022] and finance[Ghysels et al., 2005], only considering the mean of returns is insufficient. It is necessary to take risk and uncertainties into consideration. Distributional reinforcement learning (DRL) Morimura et al. [2010], Bellemare et al. [2017, 2023] addresses such issues by modeling the complete distribution of returns instead of their expectations.

In the field of DRL, one of the most fundamental tasks is to estimate the return distribution $\eta^\pi$ for a given policy $\pi$, which is referred to as distributional policy evaluation. Distributional temporal difference learning (TD) is probably the most widely-used approach for solving the distributional policy evaluation problem. A key aspect of implementing a distributional TD algorithm is how to represent the return distribution, an infinite-dimensional object, via a computationally feasible finite-dimensional parametrization. This has led to the development of two special instances of distributional TD: categorical temporal difference learning (CTD) [Bellemare et al., 2017] and quantile temporal difference learning (QTD) [Dabney et al., 2018]. These algorithms provide computationally tractable parametrizations and updating schemes of the return distribution.

[*]School of Mathematical Sciences, Peking University; email: `pengyang@pku.edu.cn`.

[†]School of Statistics and Management, Shanghai University of Finance and Economics; email: `zhangliangyu@sufe.edu.cn`.

[‡]School of Mathematical Sciences, Peking University; email: `zhzhang@math.pku.edu.cn`.

38th Conference on Neural Information Processing Systems (NeurIPS 2024).

Previous theoretical works have primarily focused on the asymptotic behaviors of distributional TD. In particular, Rowland et al. [2018] and Rowland et al. [2024a] showed the asymptotic convergences of CTD and QTD in the tabular case, respectively. A natural question arises: *can we depict the statistical efficiency of distributional TD by non-asymptotic results similar to the classic TD algorithm [Li et al., 2024]?*

## 1.1 Contributions

In this paper, we manage to answer the above question affirmatively in the synchronous setting [Kakade, 2003, Kearns et al., 2002]. Firstly, we introduce non-parametric distributional temporal difference learning (NTD) in Section 3, which is not practical but aids theoretical understanding. We show that $\widetilde{O}\left(\frac{1}{\varepsilon^{2p}(1-\gamma)^{2p+1}}\right)$ [4] iterations are sufficient to yield an estimator $\hat{\eta}^\pi$, such that the $p$-Wasserstein metric between $\hat{\eta}^\pi$ and $\eta^\pi$ is less than $\varepsilon$ with high probability (Theorem 4.1). This bound is minimax optimal (Theorem B.1) in the 1-Wasserstein metric case, if we neglect all logarithmic terms. Next, we revisit the more practical CTD, and show that, in terms of the $p$-Wasserstein metric, CTD and NTD have the same non-asymptotic convergence bounds (Theorem 4.2). It is worth pointing out that to attain such tight bounds in Theorem 4.1, we establish a Freedman's inequality in Hilbert spaces (Theorem A.2). We would believe it is of independent interest beyond the current work.

## 1.2 Related Work

**Non-asymptotic results of DRL.** Recently, there has been an emergence of work focusing on finite-sample/iteration results of the distributional policy evaluations.

Wu et al. [2023] studied the offline distributional policy evaluation problem. They solved the problem via fitted likelihood estimation (FLE) inspired by the classic offline policy evaluation algorithm fitted Q evaluation (FQE), and provided a generalization bound in the $p$-Wasserstein metric case.

Zhang et al. [2023] proposed to solve distributional policy evaluation by the model-based approach and derived corresponding sample complexity bounds, namely $\widetilde{O}\left(\frac{1}{\varepsilon^{2p}(1-\gamma)^{2p+2}}\right)$ in the $p$-Wasserstein metric case, and $\widetilde{O}\left(\frac{1}{\varepsilon^{2}(1-\gamma)^{4}}\right)$ in both the Kolmogorov-Smirnov metric and total variation metric under different conditions. Rowland et al. [2024b] proposed direct categorical fixed-point computation (DCFP), a model-based version of CTD, in which they constructed the estimator by solving a linear system directly instead of performing an iterative algorithm. They showed that the sample complexity of DCFP is $\widetilde{O}\left(\frac{1}{\varepsilon^{2}(1-\gamma)^{3}}\right)$ in the 1-Wasserstein metric case by introducing the novel stochastic categorical CDF Bellman operator and equation. Their result matches the minimax lower bound (up to logarithmic factors) $\widetilde{\Omega}\left(\frac{1}{\varepsilon^{2}(1-\gamma)^{3}}\right)$ proposed in [Zhang et al., 2023], which implies that learning the full return distribution can be as sample-efficient as learning just its expectation. It's worth noting that the algorithms analyzed in both [Zhang et al., 2023] and [Rowland et al., 2024b] are model-based, hence they are less similar to practical algorithms. While distributional TD analyzed in this paper, as a model-free method, is more practical, and also involves a more complicated theoretical analysis.

Böck and Heitzinger [2022] also considered model-free method. They proposed speedy categorical policy evaluation (SCPE), which can be regarded as CTD with an additional acceleration term. They showed that the sample complexity of SCPE is $\widetilde{O}\left(\frac{1}{\varepsilon^{2}(1-\gamma)^{4}}\right)$ in the 1-Wasserstein metric case. Compared to [Böck and Heitzinger, 2022], our work shows that even if we do not introduce any acceleration techniques to the original CTD algorithm, it is still possible to attain the near-minimax optimal sample complexity bounds. Thus, we give *sharper* bounds based on a *simpler* algorithm.

Table 1 gives more detailed comparisons of sample complexity with the previous work in the 1-Wasserstein metric. Note that solving distributional policy evaluation can also address the traditional

---

[4]Throughout this paper, the notation $f(\cdot) = \tilde{O}(g(\cdot))$ ($f(\cdot) = \tilde{\Omega}(g(\cdot))$) means that $f(\cdot)$ is order-wise no larger (smaller) than $g(\cdot)$, ignoring logarithmic factors $\mathrm{poly}(\log|\mathcal{S}|, \log|\mathcal{A}|, \log(\frac{1}{1-\gamma}), \log(\frac{1}{\varepsilon}), \log(\frac{1}{\delta}))$, as $|\mathcal{S}|, |\mathcal{A}|, \frac{1}{1-\gamma}, \frac{1}{\varepsilon}, \frac{1}{\delta} \to \infty$.

| | Sample Complexity | Algorithms | Task |
|---|---|---|---|
| [Gheshlaghi Azar et al., 2013] | $\widetilde{O}\left(\frac{1}{\varepsilon^2(1-\gamma)^3}\right)$ | Model-based | PE |
| [Li et al., 2024] | $\widetilde{O}\left(\frac{1}{\varepsilon^2(1-\gamma)^3}\right)$ | TD (Model-free) | PE |
| [Rowland et al., 2018] | Asymptotic | CTD (Model-free) | DPE |
| [Rowland et al., 2024a] | Asymptotic | QTD (Model-free) | DPE |
| [Rowland et al., 2024b] | $\widetilde{O}\left(\frac{1}{\varepsilon^2(1-\gamma)^3}\right)$ | DCFP (Model-based) | DPE |
| [Böck and Heitzinger, 2022] | $\widetilde{O}\left(\frac{1}{\varepsilon^2(1-\gamma)^4}\right)$ | SCPE (Model-free) | DPE |
| Our Work | $\widetilde{O}\left(\frac{1}{\varepsilon^2(1-\gamma)^3}\right)$ | CTD (Model-free) | DPE |

**Table 1.** Sample complexity of algorithms for solving policy evaluation (PE) in the $\ell_\infty$ norm, and distributional policy evaluation (DPE) in the supreme 1-Wasserstein metric.

policy evaluation task by taking expectation of the return distribution estimator. And the supreme 1-Wasserstein metric error of the return distribution estimator is not smaller than the $\ell_\infty$ error of the induced value function estimator (see the proof of Theorem B.1 in Appendix B), we have also listed the sample complexity of the policy evaluation task in Table 1 for comparison.

**Freedman's inequality.** Freedman's inequality was originally proposed in [Freedman, 1975]. It can be viewed as a Bernstein's inequality for martingales, which is crucial for analyzing stochastic approximation algorithms. Tropp [2011] generalized Freedman's inequality to matrix martingales. And Talebi et al. [2022] established Freedman inequalities for martingales in the setting of noncommutative probability spaces. The closest literature to ours is [Tarres and Yao, 2014] and [Martinez-Taboada and Ramdas, 2024], where they provided a special case of our Theorem A.2 with $H = 0$ independently. When $H = 0$, we can only utilize the deterministic upper bound on the quadratic variation rather than the high-probability upper bound. In certain problems, such as the distributional TD learning we aim to investigate, it is impossible to achieve the optimal upper bound using the $H = 0$ version. [Martinez-Taboada and Ramdas, 2024] also proposed an empirical Freedman's inequality in $(2, D)$-smooth Banach space, which can be used to construct confidence sets or perform hypothesis testing. To the best of our knowledge, we are the first to present this version (Theorem A.2) of Freedman's inequality in Hilbert spaces[5].

The remainder of this paper is organized as follows. In Section 2, we introduce some background of DRL and state Freedman's inequality in Hilbert spaces. In Section 3, we revisit distributional TD and propose NTD for further theoretical analysis. In Section 4, we analyze the non-asymptotic convergence bounds of NTD and CTD. Section 5 presents proof outlines of our theoretical results, and Section 6 concludes our work. We put the detailed results with Freedman's inequality in Hilbert spaces in Appendix A, and the minimax lower bound of the distributional policy evaluation task in Appendix B.

## 2  Background

An infinite-horizon tabular Markov decision process (MDP) is defined by a 5-tuple $M = \langle \mathcal{S}, \mathcal{A}, \mathcal{P}_R, P, \gamma \rangle$, where $\mathcal{S}$ represents a finite state space, $\mathcal{A}$ a finite action space, $\mathcal{P}_R$ the distribution of rewards, $P$ the transition dynamics, *i.e.*, $\mathcal{P}_R(\cdot|s, a) \in \Delta([0, 1])$, $P(\cdot|s, a) \in \Delta(\mathcal{S})$ for any state action pair $(s, a) \in \mathcal{S} \times \mathcal{A}$, and $\gamma \in (0, 1)$ a discount factor. Here we use $\Delta(\cdot)$ to represent the set of all probability distributions over some set. Given a policy $\pi \colon \mathcal{S} \to \Delta(\mathcal{A})$ and an initial state $s_0 = s \in \mathcal{S}$, a random trajectory $\{(s_t, a_t, t_t)_{t=0}^\infty\}$ can be sampled from $M$: $a_t \mid s_t \sim \pi(\cdot \mid s_t)$, $r_t \mid (s_t, a_t) \sim \mathcal{P}_R(\cdot \mid s_t, a_t)$, $s_{t+1} \mid (s_t, a_t) \sim P(\cdot \mid s_t, a_t)$ for any $t \in \mathbb{N}$. Given a trajectory, we define the return by $G^\pi(s) := \sum_{t=0}^\infty \gamma^t r_t \in \left[0, \frac{1}{1-\gamma}\right]$. We denote return distribution $\eta^\pi(s)$ as the probability distribution of $G^\pi(s)$, and $\eta^\pi := (\eta^\pi(s))_{s \in \mathcal{S}}$. The expected return $V^\pi(s) = \mathbb{E}G^\pi(s)$ is the value function in the traditional RL setting.

---

[5]In this paper, we assume that all the Hilbert spaces we encounter are separable, which can avoid measurability issues, ensure that the expectation can be defined, and guarantee tightness of any distribution. See Pisier [2016] for more details about probability in Hilbert space

## 2.1 Distributional Bellman Equation and Operator

Recall that the classic policy evaluation aims at computing the value functions $V^\pi$. It is known that $V^\pi = (V^\pi(s))_{s \in \mathcal{S}}$ satisfy the Bellman equation. That is, for any $s \in \mathcal{S}$,

$$V^\pi(s) = [T^\pi(V^\pi)](s) = \mathbb{E}_{a \sim \pi(\cdot|s), r \sim \mathcal{P}_R(\cdot|s,a), s' \sim P(\cdot|s,a)} [r + \gamma V^\pi(s')]. \tag{1}$$

The operator $T^\pi \colon \mathbb{R}^\mathcal{S} \to \mathbb{R}^\mathcal{S}$ is called the Bellman operator, and $V^\pi$ is a fixed point of $T^\pi$.

The task of distribution policy evaluation is finding $\eta^\pi$ given some fixed policy $\pi$. $\eta^\pi$ satisfies a distributional version of the Bellman equation (1). That is, for any $s \in \mathcal{S}$,

$$\eta^\pi(s) = (\mathcal{T}^\pi \eta^\pi)(s) = \mathbb{E}_{a \sim \pi(\cdot|s), r \sim \mathcal{P}_R(\cdot|s,a), s' \sim P(\cdot|s,a)} \left[ (b_{r,\gamma})_\# \, \eta^\pi(s') \right], \tag{2}$$

where $b_{r,\gamma} \colon \mathbb{R} \to \mathbb{R}$ is an affine function defined by $b_{r,\gamma}(x) = r + \gamma x$. And $f_\# \mu$ is the push forward measure of $\mu$ through any function $f \colon \mathbb{R} \to \mathbb{R}$, so that $f_\# \mu(A) = \mu(f^{-1}(A))$ for any Borel set $A$, where $f^{-1}(A) := \{x \colon f(x) \in A\}$. The operator $\mathcal{T}^\pi \colon \Delta\left(\left[0, \frac{1}{1-\gamma}\right]\right)^\mathcal{S} \to \Delta\left(\left[0, \frac{1}{1-\gamma}\right]\right)^\mathcal{S}$ is known as the distributional Bellman operator, and $\eta^\pi$ is a fixed point of $\mathcal{T}^\pi$. For notational simplicity, we denote $\Delta\left(\left[0, \frac{1}{1-\gamma}\right]\right)$ as $\mathscr{P}$ from now on.

## 2.2 $\mathcal{T}^\pi$ as Contraction in $\mathscr{P}$

A key property of the Bellman operator $T^\pi$ is that it is a $\gamma$-contraction w.r.t. the supreme norm (*i.e.* $\ell_\infty$ norm). However, before we can properly discuss the contraction properties of $\mathcal{T}^\pi$, we need to specify a metric $d$ on $\mathscr{P}$. And for any metric $d$ on $\mathscr{P}$, we denote $\bar{d}$ as the corresponding supreme metric on $\mathscr{P}^\mathcal{S}$, *i.e.*, $\bar{d}(\eta, \eta') := \max_{s \in \mathcal{S}} d(\eta(s), \eta'(s))$ for any $\eta, \eta' \in \mathscr{P}^\mathcal{S}$.

Suppose $\mu$ and $\nu$ are two probability distributions on $\mathbb{R}$ with finite $p$-moments for $p \in [1, \infty]$. The $p$-Wasserstein metric between $\mu$ and $\nu$ is defined as $W_p(\mu, \nu) := \left( \inf_{\kappa \in \Gamma(\mu,\nu)} \int_{\mathbb{R}^2} |x - y|^p \, \kappa(dx, dy) \right)^{1/p}$. Each element $\kappa \in \Gamma(\mu, \nu)$ is a coupling of $\mu$ and $\nu$, *i.e.*, a joint distribution on $\mathbb{R}^2$ with prescribed marginals $\mu$ and $\nu$ on each "axis." When $p = 1$ we have $W_1(\mu, \nu) = \int_{\mathbb{R}} |F_\mu(x) - F_\nu(x)| dx$, where $F_\mu$ and $F_\nu$ are the cumulative distribution function of $\mu$ and $\nu$, respectively. It is shown that $\mathcal{T}^\pi$ is a $\gamma$-contraction w.r.t. the supreme $p$-Wasserstein metric $\bar{W}_p$.

**Proposition 2.1.** *[Bellemare et al., 2023, Propositions 4.15] The distributional Bellman operator is a $\gamma$-contraction on $\mathscr{P}^\mathcal{S}$ w.r.t. the supreme $p$-Wasserstein metric for $p \in [1, \infty]$. That is, for any $\eta, \eta' \in \mathscr{P}^\mathcal{S}$, we have $\bar{W}_p\left(\mathcal{T}^\pi \eta, \mathcal{T}^\pi \eta'\right) \leq \gamma \bar{W}_p(\eta, \eta')$.*

The $\ell_p$ metric between $\mu$ and $\nu$ is defined as $\ell_p(\mu, \nu) = \left( \int_{\mathbb{R}} |F_\mu(x) - F_\nu(x)|^p \, dx \right)^{\frac{1}{p}}$ for $p \in [1, \infty)$, and $\mathcal{T}^\pi$ is $\gamma^{\frac{1}{p}}$-contraction w.r.t. the supreme $\ell_p$ metric $\bar{\ell}_p$.

**Proposition 2.2.** *[Bellemare et al., 2023, Propositions 4.20] The distributional Bellman operator is a $\gamma^{\frac{1}{p}}$-contraction on $\mathscr{P}^\mathcal{S}$ w.r.t. the supreme $\ell_p$ metric for $p \in [1, \infty)$. That is, for any $\eta, \eta' \in \mathscr{P}^\mathcal{S}$, we have $\bar{\ell}_p\left(\mathcal{T}^\pi \eta, \mathcal{T}^\pi \eta'\right) \leq \gamma^{\frac{1}{p}} \bar{\ell}_p(\eta, \eta')$.*

Note that the $\ell_1$ metric coincides with the 1-Wasserstein metric. And the $\ell_2$ metric is also called the Cramér metric, which plays an important role in subsequent analysis because the zero-mass signed measure space equipped with this metric $\left(\mathcal{M}, \|\cdot\|_{\ell_2}\right)$ (defined in Section 5.1) is a Hilbert space[6]. Thereby, we can apply Freedman's inequality in Hilbert spaces.

## 2.3 Freedman's Inequality in Hilbert Spaces

Just as Freedman's inequality is essential for the theory of TD (Theorem 1 in [Li et al., 2024]), a Hilbert space version of Freedman's inequality is indispensable for deriving the minimax non-asymptotic convergence bound for distributional TD. At the moment, we state a Hilbert space version of the original Freedman's inequality (Theorem 1.6 in [Freedman, 1975]), and more detailed results can be found in Appendix A.

---

[6]In fact, the space $\left(\mathcal{M}, \|\cdot\|_{\ell_2}\right)$ is not complete. However, the completeness property does not affect the non-asymptotic analysis, see Section 5.1 for more details.

Let $\mathcal{X}$ be a Hilbert space, $\{X_i\}_{i=1}^n$ be an $\mathcal{X}$-valued martingale difference sequence adapted to the filtration $\{\mathcal{F}_i\}_{i=1}^n$, $Y_i := \sum_{j=1}^i X_j$ be the corresponding martingale, and $W_i := \sum_{j=1}^i \sigma_j^2$ be the corresponding quadratic variation process. Here $\sigma_j^2 := \mathbb{E}_{j-1}\|X_j\|^2$, and $\mathbb{E}_i[\cdot] := \mathbb{E}[\cdot|\mathcal{F}_i]$ denotes the conditional expectation.

**Theorem 2.1** (Freedman's inequality in Hilbert spaces). *Suppose* $\max_{i\in[n]}\|X_i\| \leq b$ *for some constant* $b > 0$. *Then, for any* $\varepsilon$ *and* $\sigma > 0$*, the following inequality holds*

$$\mathbb{P}\left(\exists k \in [n], s.t. \|Y_k\| \geq \varepsilon \text{ and } W_k \leq \sigma^2\right) \leq 2\exp\left\{-\frac{\varepsilon^2/2}{\sigma^2 + b\varepsilon/3}\right\}.$$

# 3 Distributional Temporal Difference Learning

If the MDP $M = \langle \mathcal{S}, \mathcal{A}, \mathcal{P}_R, P, \gamma \rangle$ is known, and because $V^\pi$ is the fixed point of the contraction $T^\pi$, $V^\pi$ can be evaluated via the famous dynamic programming (DP) algorithm. To be concrete, for any initialization $V^{(0)} \in \mathbb{R}^{\mathcal{S}}$, if we define the iteration sequence $V^{(k+1)} = T^\pi(V^{(k)})$ for $k \in \mathbb{N}$, we have $\lim_{k\to\infty}\|V^{(k)} - V^\pi\|_\infty = 0$ by the contraction mapping theorem (Proposition 4.7 in [Bellemare et al., 2023]).

Similarly, the distributional dynamic programming algorithm defines the iteration sequence as $\eta^{(k+1)} = \mathcal{T}^\pi \eta^{(k)}$ for any initialization $\eta^{(0)}$. In the same way, we have $\lim_{k\to\infty} \bar{W}_p(\eta^{(k)}, \eta^\pi) = 0$ for $p \in [1, \infty]$ and $\lim_{k\to\infty} \bar{\ell}_p(\eta^{(k)}, \eta^\pi) = 0$ for $p \in [1, \infty)$.

In most application scenarios, the transition dynamic $P$ and reward distribution $\mathcal{P}_R$ are unknown, and instead we can only get samples of $P$ and $\mathcal{P}_R$ in a streaming manner. In this paper, we assume a generative model [Kakade, 2003, Kearns et al., 2002] is accessible, which generates independent samples for all states in each iteration, *i.e.*, in the $t$-th iteration, we collect sample $a_t(s) \sim \pi(\cdot|s), s_t(s) \sim P(\cdot|s, a_t(s)), r_t(s) \sim \mathcal{P}_R(\cdot|s, a_t(s))$ for each $s \in \mathcal{S}$. Similar to TD [Sutton, 1988] in classic RL, distributional TD also employs the stochastic approximation (SA) [Robbins and Monro, 1951] technique to address the aforementioned problem and can be viewed as an approximate version of distributional DP.

**Non-parametric Distributional TD** We first introduce non-parametric distributional temporal difference learning (NTD), which is helpful in the theoretical understanding of distributional TD. In the setting of NTD, we assume the return distributions can be precisely updated without any parametrization. For any initialization $\eta_0^\pi \in \mathscr{P}^{\mathcal{S}}$, the updating scheme is given by

$$\eta_t^\pi = (1 - \alpha_t)\eta_{t-1}^\pi + \alpha_t \mathcal{T}_t^\pi \eta_{t-1}^\pi$$

for any $t \geq 1$. Here $\alpha_t$ is the step size. The empirical Bellman operator at the $t$-th iteration $\mathcal{T}_t^\pi$ is defined as

$$(\mathcal{T}_t^\pi \eta)(s) = (b_{r_t(s),\gamma})_\# (\eta(s_{t+1})),$$

which is an unbiased estimator of $(\mathcal{T}^\pi \eta)(s)$. It is evident that NTD is a SA modification of distributional DP. Consequently, we can analyze NTD using the techniques from the SA area.

**Categorical Distributional TD** Now, we revisit the more practical CTD. In this case, the updates in CTD is computationally tractable, due to the following categorical parametrization of probability distributions:

$$\mathscr{P}_K := \left\{\sum_{k=0}^K p_k \delta_{x_k} : p_0, \ldots, p_K \geq 0, \sum_{k=0}^K p_k = 1\right\},$$

where $K \in \mathbb{N}$, and $0 \leq x_0 < \cdots < x_K \leq \frac{1}{1-\gamma}$ are fixed points of the support. For simplicity, we assume $\{x_k\}_{k=0}^K$ are equally-spaced, *i.e.*, $x_k = \frac{k}{K(1-\gamma)}$. We denote the gap between two points by $\iota_K = \frac{1}{K(1-\gamma)}$. When updating the return distributions, we need to evaluate the $\ell_2$-projection of $\mathscr{P}_K$, $\Pi_K: \mathscr{P} \to \mathscr{P}_K$, $\Pi_K \mu := \operatorname{argmin}_{\hat{\mu}\in\mathscr{P}_K} \ell_2(\mu, \hat{\mu})$. It can be shown (Proposition 5.14 in [Bellemare et al., 2023]) that the projection is uniquely given by

$$\Pi_K \mu = \sum_{k=0}^K p_k(\mu)\delta_{x_k}, \text{ where } p_k(\mu) = \mathbb{E}_{X\sim\mu}\left[\left(1 - \left|\frac{X - x_k}{\iota_K}\right|\right)_+\right],$$

$(x)_+ := \max\{x, 0\}$ for any $x \in \mathbb{R}$. It is known that $\Pi_K$ is non-expansive w.r.t. the Cramér metric (Lemma 5.23 in [Bellemare et al., 2023]), *i.e.*, $\ell_2(\Pi_K \mu, \Pi_K \nu) \le \ell_2(\mu, \nu)$ for any $\mu, \nu \in \mathscr{P}$. For any $\eta \in \mathscr{P}^{\mathcal{S}}$, $s \in \mathcal{S}$, we slightly abuse the notation and define $(\Pi_K \eta)(s) := \Pi_K \eta(s)$. $\Pi_K$ is still non-expansive w.r.t. $\bar{\ell}_2$. Hence $\mathcal{T}^{\pi,K} := \Pi_K \mathcal{T}^\pi$ is a $\sqrt{\gamma}$-contraction w.r.t. $\bar{\ell}_2$, we denote its unique fixed point as $\eta^{\pi,K} \in \mathscr{P}_K^{\mathcal{S}}$. The approximation error induced by categorical parametrization is given by (Proposition 3 in Rowland et al. [2018])

$$\bar{\ell}_2(\eta^\pi, \eta^{\pi,K}) \le \frac{1}{\sqrt{K}(1-\gamma)}, \quad \bar{W}_1(\eta^\pi, \eta^{\pi,K}) \le \frac{1}{\sqrt{1-\gamma}} \bar{\ell}_2(\eta^\pi, \eta^{\pi,K}) \le \frac{1}{\sqrt{K}(1-\gamma)^{3/2}}. \quad (3)$$

Now, we are ready to give the updating scheme of CTD, given any initialization $\eta_0^\pi \in \mathscr{P}_K^{\mathcal{S}}$,

$$\eta_t^\pi = (1-\alpha_t)\eta_{t-1}^\pi + \alpha_t \Pi_K \mathcal{T}_t^\pi \eta_{t-1}^\pi$$

for any $t \ge 1$. We can find that the only difference between CTD and NTD lies in the additional application of the projection operator $\Pi_K$ at each iteration in CTD.

## 4 Statistical Analysis

In this section, we state our main results. For both NTD and CTD, we give the non-asymptotic convergence rates of $\bar{W}_p(\eta_T^\pi, \eta^\pi)$ and $\bar{\ell}_2(\eta_T^\pi, \eta^\pi)$, respectively.

### 4.1 Non-asymptotic Analysis of NTD

We first provide a non-asymptotic convergence rate of $\bar{W}_1(\eta_T^\pi, \eta^\pi)$ for NTD, which is minimax optimal (Theorem B.1) up to logarithmic factors.

**Theorem 4.1** (Sample complexity of NTD in the 1-Wasserstein metric). *Given any $\delta \in (0,1)$ and $\varepsilon \in (0,1)$, let the initialization be $\eta_0^\pi \in \mathscr{P}^{\mathcal{S}}$, the total update number $T$ satisfy*

$$T \ge \frac{C_1 \log^3 T}{\varepsilon^2(1-\gamma)^3} \log \frac{|\mathcal{S}|T}{\delta}$$

*for some large universal constant $C_1 > 1$, i.e., $T = \widetilde{O}\left(\frac{1}{\varepsilon^2(1-\gamma)^3}\right)$, and the step size $\alpha_t$ satisfy*

$$\frac{1}{1 + \frac{c_2(1-\sqrt{\gamma})t}{\log t}} \le \alpha_t \le \frac{1}{1 + \frac{c_3(1-\sqrt{\gamma})t}{\log t}}$$

*for some small universal constants $c_2 > c_3 > 0$. Then, with probability at least $1-\delta$, the last iterate estimator satisfies $\bar{W}_1(\eta_T^\pi, \eta^\pi) \le \varepsilon$.*

Because $\bar{W}_1(\eta_T^\pi, \eta^\pi) \le \frac{1}{1-\gamma}$ always holds, we can translate the high probability bound to a mean error bound, that is,

$$\mathbb{E}\left[\bar{W}_1(\eta_T^\pi, \eta^\pi)\right] \le \varepsilon(1-\delta) + \frac{\delta}{1-\gamma} \le 2\varepsilon$$

if we take $\delta \le \varepsilon(1-\gamma)$. In the subsequent discussion, we will not state the mean error bound conclusions for the sake of brevity.

The key idea of our proof is to first expand the error term $\bar{W}_1(\eta_T^\pi, \eta^\pi)$ over the time steps. Then it can be decomposed into an initial error term and a martingale term. The initial error term becomes smaller as the iteration goes due to the contraction properties of $\mathcal{T}^\pi$. To control the martingale term, we first use the basic inequality (Lemma E.1) $W_1(\mu, \nu) \le \frac{1}{\sqrt{1-\gamma}} \ell_2(\mu, \nu)$, which allows us to analyze this error term in the Hilbert space $(\mathcal{M}, \|\cdot\|_{\ell_2})$ defined in Section 5.1. Consequently, we can bound it using Freedman's inequality in the Hilbert space (Theorem A.2). A more detailed outline of proof can be found in Section 5.2.

Combining Theorem 4.1 with the basic inequality $\bar{W}_p(\eta, \eta') \le \frac{1}{(1-\gamma)^{1-\frac{1}{p}}} \bar{W}_1^{\frac{1}{p}}(\eta, \eta')$ for any $\eta, \eta' \in \mathscr{P}^{\mathcal{S}}$ (Lemma E.1), we can derive that $T = \widetilde{O}\left(\frac{1}{\varepsilon^{2p}(1-\gamma)^{2p+1}}\right)$ iterations are sufficient to ensure

$\bar{W}_p(\eta_T^\pi, \eta^\pi) \leq \varepsilon$. As pointed out in the example after Corollary 3.1 in [Zhang et al., 2023], when $p > 1$, the slow rate in terms of $\varepsilon$ is inevitable without additional regularity conditions.

Although the 1-Wasserstein metric cannot bound the Cramér metric properly, by making slight modifications to the proof we have the following non-asymptotic convergence rate of $\bar{\ell}_2(\eta_T^\pi, \eta^\pi)$. See Appendix C.5 for our proof.

**Corollary 4.1** (Sample complexity of NTD in the Cramér metric). *Given any $\delta \in (0, 1)$ and $\varepsilon \in (0, 1)$, let the initial value $\eta_0^\pi \in \mathscr{P}^\mathcal{S}$, the total update number $T$ satisfy*

$$T \geq \frac{C_1 \log^3 T}{\varepsilon^2 (1 - \gamma)^{5/2}} \log \frac{|\mathcal{S}| T}{\delta}$$

*for some large universal constant $C_1 > 1$, i.e., $T = \widetilde{O}\left(\frac{1}{\varepsilon^2(1-\gamma)^{5/2}}\right)$, and the step size $\alpha_t$ satisfy*

$$\frac{1}{1 + \frac{c_2(1-\sqrt{\gamma})t}{\log t}} \leq \alpha_t \leq \frac{1}{1 + \frac{c_3(1-\sqrt{\gamma})t}{\log t}}$$

*for some small universal constants $c_2 > c_3 > 0$. Then, with probability at least $1 - \delta$, the last iterate estimator satisfies $\bar{\ell}_2\left(\eta_T^\pi, \eta^\pi\right) \leq \varepsilon$.*

## 4.2 Non-asymptotic Analysis of CTD

We first state a parallel result to Theorem 4.1.

**Theorem 4.2** (Sample complexity of CTD in the 1-Wasserstein metric). *Given any $\delta \in (0, 1)$ and $\varepsilon \in (0, 1)$, suppose $K > \frac{4}{1-\gamma}$, the initial value $\eta_0^\pi \in \mathscr{P}_K^\mathcal{S}$, the total update number $T$ satisfies*

$$T \geq \frac{C_1 \log^3 T}{\varepsilon^2 (1 - \gamma)^3} \log \frac{|\mathcal{S}| T}{\delta}$$

*for some large universal constant $C_1 > 1$, i.e., $T = \widetilde{O}\left(\frac{1}{\varepsilon^2(1-\gamma)^3}\right)$, and the step size $\alpha_t$ satisfies*

$$\frac{1}{1 + \frac{c_2(1-\sqrt{\gamma})t}{\log t}} \leq \alpha_t \leq \frac{1}{1 + \frac{c_3(1-\sqrt{\gamma})t}{\log t}}$$

*for some small universal constants $c_2 > c_3 > 0$. Then, with probability at least $1 - \delta$, the last iterate estimator satisfies $\bar{W}_1\left(\eta_T^\pi, \eta^{\pi,K}\right) \leq \frac{\varepsilon}{2}$. Furthermore, according to the upper bound (3) of the approximation error $\bar{W}_1\left(\eta^{\pi,K}, \eta^\pi\right)$, if we take $K > \frac{4}{\varepsilon^2(1-\gamma)^3}$, we have $\bar{W}_1\left(\eta_T^\pi, \eta^\pi\right) \leq \varepsilon$.*

Note that the order (modulo logarithmic factors) of sample complexity of CTD is better than the previous results of SCPE [Böck and Heitzinger, 2022], and we do not need the additional term introduced in the updating scheme of SCPE.

The proof of this theorem is almost the same as that of Theorem 4.1, we outline the proof in Section 5.2. The $\bar{W}_1$ metric result can be translated into sample complexity bound $\widetilde{O}\left(\frac{1}{\varepsilon^{2p}(1-\gamma)^{2p+1}}\right)$ in the $\bar{W}_p$ metric. We comment that this theoretical result matches the sample complexity bound in the model-based setting [Rowland et al., 2024b].

As in the NTD setting, we have the following non-asymptotic convergence rate of $\bar{\ell}_2(\eta_T^\pi, \eta^\pi)$ as a corollary of Theorem 4.2. See Appendix C.5 for the proof.

**Corollary 4.2** (Sample complexity of CTD in the Cramér metric). *For any given $\delta \in (0, 1)$ and $\varepsilon \in (0, 1)$, suppose $K > \frac{4}{1-\gamma}$, the initialization is $\eta_0^\pi \in \mathscr{P}_K^\mathcal{S}$, the total update number $T$ satisfies*

$$T \geq \frac{C_1 \log^3 T}{\varepsilon^2 (1 - \gamma)^{5/2}} \log \frac{|\mathcal{S}| T}{\delta}$$

*for some large universal constant $C_1 > 1$, i.e., $T = \widetilde{O}\left(\frac{1}{\varepsilon^2(1-\gamma)^{5/2}}\right)$, and the step size $\alpha_t$ satisfies*

$$\frac{1}{1 + \frac{c_2(1-\sqrt{\gamma})t}{\log t}} \leq \alpha_t \leq \frac{1}{1 + \frac{c_3(1-\sqrt{\gamma})t}{\log t}}$$

*for some small universal constants $c_2 > c_3 > 0$. Then, with probability at least $1 - \delta$, the last iterate estimator satisfies $\bar{\ell}_2\left(\eta_T^\pi, \eta^{\pi,K}\right) \leq \frac{\varepsilon}{2}$. Furthermore, according to the upper bound (3) of the approximation error $\bar{\ell}_2\left(\eta^{\pi,K}, \eta^\pi\right)$, if we take $K > \frac{4}{\varepsilon^2(1-\gamma)^2}$, we have $\bar{\ell}_2\left(\eta_T^\pi, \eta^\pi\right) \leq \varepsilon$.*

# 5 Proof Outlines

In this section, we will outline the proofs of our main theoretical results (Theorem 4.1, Corollary 4.1, Theorem 4.2, and Corollary 4.2). Before diving into the details of the proofs, we first define some notation.

## 5.1 Zero-mass Signed Measure Space

To analyze the distance between the estimator and the ground-truth $\eta^\pi$, we will work with the zero-mass signed measure space $\mathcal{M}$ defined as follows

$$\mathcal{M} := \left\{ \mu \colon \mu \text{ is a signed measure with } |\mu|(\mathbb{R}) < \infty, \mu(\mathbb{R}) = 0, \mathrm{supp}(\mu) \subseteq [0, \frac{1}{1-\gamma}] \right\},$$

where $|\mu|$ is the total variation measure of $\mu$, and $\mathrm{supp}(\mu)$ is the support of $\mu$. See [Bogachev, 2007] for more details about signed measures.

For any $\mu \in \mathcal{M}$, we define its cumulative function as $F_\mu(x) := \mu[0,x)$. We can check that $F_\mu$ is linear w.r.t. $\mu$, that is, $F_{\alpha\mu+\beta\nu} = \alpha F_\mu + \beta F_\nu$ for any $\alpha, \beta \in \mathbb{R}, \mu, \nu \in \mathcal{M}$.

To analyze the Cramér metric case, we define the following Cramér inner product on $\mathcal{M}$:

$$\langle \mu, \nu \rangle_{\ell_2} := \int_0^{\frac{1}{1-\gamma}} F_\mu(x) F_\nu(x) dx.$$

It is easy to verify that $\langle \cdot, \cdot \rangle_{\ell_2}$ is indeed an inner product on $\mathcal{M}$. The corresponding norm, called the Cramér norm, is given by $\|\mu\|_{\ell_2} = \sqrt{\langle \mu, \mu \rangle_{\ell_2}} = \sqrt{\int_0^{\frac{1}{1-\gamma}} (F_\mu(x))^2 dx}$. We have $\nu_1 - \nu_2 \in \mathcal{M}$ and $\|\nu_1 - \nu_2\|_{\ell_2} = \ell_2(\nu_1, \nu_2)$ for any $\nu_1, \nu_2 \in \mathscr{P}$.

The $W_1$ norm on $\mathcal{M}$ is defined as $\|\mu\|_{W_1} := \int_0^{\frac{1}{1-\gamma}} |F_\mu(x)| dx$. We have $\|\nu_1 - \nu_2\|_{W_1} = W_1(\nu_1, \nu_2)$ for any $\nu_1, \nu_2 \in \mathscr{P}$.

We can extend the distributional Bellman operator $\mathcal{T}^\pi$ and the Cramér projection operator $\Pi_K$ naturally to $\mathcal{M}^{\mathcal{S}}$. Here, the product space $\mathcal{M}^{\mathcal{S}}$ is also a Banach space, and we use the supreme norm: $\|\eta\|_{\bar{\ell}_2} := \max_{s \in \mathcal{S}} \|\eta(s)\|_{\ell_2}$, and $\|\eta\|_{\bar{W}_1} := \max_{s \in \mathcal{S}} \|\eta(s)\|_{W_1}$ for any $\eta \in \mathcal{M}^{\mathcal{S}}$. We denote by $\mathcal{I}$ the identity operator in $\mathcal{M}^{\mathcal{S}}$.

When the norm $\|\cdot\|$ is applied to $A \in \mathcal{L}(\mathcal{X})$, where $\mathcal{X}$ is any Banach space, and $\mathcal{L}(\mathcal{X})$ is the space of all bounded linear operators in $\mathcal{X}$, we refer $\|A\|$ to the operator norm of $A$, which is defined as $\|A\| := \sup_{\eta \in \mathcal{X}, \|\eta\|=1} \|A\eta\|$. With this notation, $\mathcal{L}(\mathcal{X}) = \{A \colon A \text{ is a linear operator mapping from } \mathcal{X} \text{ to } \mathcal{X}, and \|A\| < \infty\}$.

**Proposition 5.1.** *$\mathcal{T}^\pi$ and $\Pi_K$ are linear operators in $\mathcal{M}^{\mathcal{S}}$. Furthermore, $\|\mathcal{T}^\pi\|_{\bar{\ell}_2} \leq \sqrt{\gamma}$, $\|\mathcal{T}^\pi\|_{\bar{W}_1} \leq \gamma$, $\|\Pi_K\|_{\bar{\ell}_2} = 1$, and $\|\Pi_K\|_{\bar{W}_1} \leq 1$.*

The proof of the last inequality can be found in the proof of Lemma C.4, while the remaining results are trivial. We omit the proofs for brevity.

Moreover, we have the following matrix (of operators) representations of $\mathcal{T}^\pi$ and $\Pi_K$: $\mathcal{T}^\pi \in \mathcal{L}(\mathcal{M})^{\mathcal{S} \times \mathcal{S}}$ for any $\eta \in \mathcal{M}^{\mathcal{S}}$,

$$(\mathcal{T}^\pi \eta)(s) = \sum_{a \in \mathcal{A}, s' \in \mathcal{S}} \pi(a \mid s) P(s' \mid s, a) \int_0^1 (b_{r,\gamma})_\# \eta(s') \mathcal{P}_R(dr \mid s, a) = \sum_{s' \in \mathcal{S}} \mathcal{T}^\pi(s, s') \eta(s'),$$

where $\mathcal{T}^\pi(s, s') \in \mathcal{L}(\mathcal{M})$ for any $\nu \in \mathcal{M}$,

$$\mathcal{T}^\pi(s, s') \nu = \sum_{a \in \mathcal{A}} \pi(a \mid s) P(s' \mid s, a) \int_0^1 (b_{r,\gamma})_\# \nu \mathcal{P}_R(dr \mid s, a).$$

It can be verified that $\|\mathcal{T}(s, s')\|_{\ell_2} \leq \sqrt{\gamma} \sum_{a \in \mathcal{A}} \pi(a \mid s) P(s' \mid s, a) =: \sqrt{\gamma} P^\pi(s'|s)$. Similarly, $\|\mathcal{T}(s, s')\|_{W_1} \leq \gamma P^\pi(s'|s)$, and $\Pi_K = \mathrm{diag}\left(\Pi_K|_{\mathcal{M}}\right)_{s \in \mathcal{S}} \in \mathcal{L}(\mathcal{M})^{\mathcal{S} \times \mathcal{S}}$. With these representations, $\Pi_K \mathcal{T}^\pi \in \mathcal{L}(\mathcal{M})^{\mathcal{S} \times \mathcal{S}}$ can be interpreted as matrix multiplication, where the scalar multiplication is replaced by the composition of operators. It can be verified that $(\Pi_K \mathcal{T}^\pi)(s, s') = \Pi_K \mathcal{T}^\pi(s, s')$, and $\|(\Pi_K \mathcal{T}^\pi)(s, s')\|_{\ell_2} \leq \sqrt{\gamma} P^\pi(s'|s)$.

**Remark 1:** In Lemma E.2, we show that both $\left(\mathcal{M}, \|\cdot\|_{\ell_2}\right)$ and $\left(\mathcal{M}, \|\cdot\|_{W_1}\right)$ are separable. And in Lemma E.3, we show that $\left(\mathcal{M}, \|\cdot\|_{W_1}\right)$ is not complete. To resolve this problem, we will use their completions to replace them without loss of generality, because the completeness property does not affect the separability. For simplicity, we still use $\mathcal{M}$ to denote the completion space. According to the BLT theorem [Theorem 5.19 Hunter and Nachtergaele, 2001], any bounded linear operator can be extended to the completion space, and still preserves its operator norm.

## 5.2 Analysis of Theorems 4.1 and 4.2

For simplicity, we abbreviate both $\|\cdot\|_{\bar{\ell}_2}$ and $\|\cdot\|_{\ell_2}$ as $\|\cdot\|$ in this part. For all $t \in [T] := \{1, 2, \cdots, T\}$, we denote $\mathcal{T}_t := \mathcal{T}_t^\pi$, $\mathcal{T} := \mathcal{T}^\pi$, $\eta := \eta^\pi$ for NTD; $\mathcal{T}_t := \Pi_K \mathcal{T}_t^\pi$, $\mathcal{T} := \Pi_K \mathcal{T}^\pi$, $\eta := \eta^{\pi,K}$ for CTD; and $\eta_t := \eta_t^\pi$, $\Delta_t := \eta_t - \eta \in \mathcal{M}^{\mathcal{S}}$ for both NTD and CTD. According to Lemma E.4, $\eta_t \in \mathscr{P}^{\mathcal{S}}$ for NTD and $\eta_t \in \mathscr{P}_K^{\mathcal{S}}$ for CTD. Our goal is to bound the $\bar{W}_1$ norm of the error term $\|\Delta_T\|_{\bar{W}_1}$. This can be achieved by bounding $\|\Delta_T\|$, as $\|\Delta_T\|_{\bar{W}_1} \leq \frac{1}{\sqrt{1-\gamma}} \|\Delta_T\|$.

According to the updating rule, we have the error decomposition

$$
\begin{aligned}
\Delta_t &= \eta_t - \eta \\
&= (1 - \alpha_t)\eta_{t-1} + \alpha_t \mathcal{T}_t \eta_{t-1} - \eta \\
&= (1 - \alpha_t)\Delta_{t-1} + \alpha_t \left(\mathcal{T}_t \eta_{t-1} - \mathcal{T}\eta\right) \\
&= (1 - \alpha_t)\Delta_{t-1} + \alpha_t \left(\mathcal{T}_t - \mathcal{T}\right)\eta_{t-1} + \alpha_t \mathcal{T}\left(\eta_{t-1} - \eta\right) \\
&= \left[(1 - \alpha_t)\mathcal{I} + \alpha_t \mathcal{T}\right]\Delta_{t-1} + \alpha_t \left(\mathcal{T}_t - \mathcal{T}\right)\eta_{t-1}.
\end{aligned}
$$

Applying it recursively, we can further decompose the error into two terms

$$
\Delta_T = \underbrace{\prod_{t=1}^{T} \left[(1 - \alpha_t)\mathcal{I} + \alpha_t \mathcal{T}\right]\Delta_0}_{\text{(I)}} + \underbrace{\sum_{t=1}^{T} \alpha_t \prod_{i=t+1}^{T} \left[(1 - \alpha_i)\mathcal{I} + \alpha_i \mathcal{T}\right]\left(\mathcal{T}_t - \mathcal{T}\right)\eta_{t-1}}_{\text{(II)}},
$$

where $\prod_{k=1}^{t} \boldsymbol{A}_k$ is defined as $\boldsymbol{A}_t \boldsymbol{A}_{t-1} \cdots \boldsymbol{A}_1$ for any operators or matrices $\{\boldsymbol{A}_k\}_{k=1}^{t}$ throughout the paper. Term (I) is an initial error term that becomes negligible when $T$ is large because $\mathcal{T}$ is a contraction. Term (II) can be bounded via Freedman's inequality in the Hilbert space (Theorem A.2). Combining the two upper bound, we can establish a recurrence relation. Solving this relation will lead to the conclusion.

We first establish the conclusion for step sizes that depend on $T$. Specifically, we consider

$$
T \geq \frac{C_4 \log^3 T}{\varepsilon^2 (1 - \gamma)^3} \log \frac{|\mathcal{S}| T}{\delta},
$$

$$
\frac{1}{1 + \frac{c_5(1 - \sqrt{\gamma})T}{\log^2 T}} \leq \alpha_t \leq \frac{1}{1 + \frac{c_6(1 - \sqrt{\gamma})t}{\log^2 T}},
$$

where $c_5 > c_6 > 0$ are small constants satisfying $c_5 c_6 \leq \frac{1}{8}$, and $C_4 > 1$ is a large constant depending only on $c_5$ and $c_6$. As shown in Appendix C.1, once we have established the conclusion in this setting, we can recover the original conclusion stated in the theorem.

Now, we introduce the following useful quantities involving step sizes and $\gamma$

$$
\beta_k^{(t)} := \begin{cases}
\prod_{i=1}^{t} \left(1 - \alpha_i(1 - \sqrt{\gamma})\right), & \text{if } k = 0, \\
\alpha_k \prod_{i=k+1}^{t} \left(1 - \alpha_i(1 - \sqrt{\gamma})\right), & \text{if } 0 < k < t, \\
\alpha_T, & \text{if } k = t.
\end{cases}
$$

The following lemma provides useful bounds for $\beta_k^{(t)}$.

**Lemma 5.1.** *Suppose $c_5 c_6 \leq \frac{1}{8}$. Then, for all $t \geq \frac{T}{c_6 \log T}$, we have that*

$$
\beta_k^{(t)} \leq \frac{1}{T^2}, \text{ for } 0 \leq k \leq \frac{t}{2}; \qquad \beta_k^{(t)} \leq \frac{2 \log^3 T}{(1 - \sqrt{\gamma})T}, \text{ for } \frac{t}{2} < k \leq t.
$$

The proof can be found in Appendix C.2. From now on, we only consider $t \geq \frac{T}{c_6 \log T}$.

The upper bound of term (I) is given by

$$\text{(I)} \leq \prod_{k=1}^{t} \|(1-\alpha_k)\mathcal{I}+\alpha_k\mathcal{T}\| \|\Delta_0\| \leq \prod_{k=1}^{t} ((1-\alpha_k)+\alpha_k\sqrt{\gamma}) \frac{1}{\sqrt{1-\gamma}} = \frac{\beta_0^{(t)}}{\sqrt{1-\gamma}} \leq \frac{1}{\sqrt{1-\gamma}T^2},$$

where $\|\Delta_0\| \leq \sqrt{\int_0^{\frac{1}{1-\gamma}} dx} = \frac{1}{\sqrt{1-\gamma}}$.

As for term (II), we have the following upper bound with high probability by utilizing Freedman's inequality (Theorem A.2).

**Lemma 5.2.** *For any $\delta \in (0,1)$, with probability at least $1-\delta$, we have for all $t \geq \frac{T}{c_6 \log T}$, in the NTD case,*

$$\left\| \sum_{k=1}^{t} \alpha_k \prod_{i=k+1}^{t} [(1-\alpha_i)\mathcal{I}+\alpha_i\mathcal{T}] (\mathcal{T}_k - \mathcal{T}) \eta_{k-1} \right\|$$

$$\leq 34 \sqrt{\frac{(\log^3 T) \left(\log \frac{|\mathcal{S}|T}{\delta}\right)}{(1-\gamma)^2 T} \left(1 + \max_{k:\, t/2 < k \leq t} \|\Delta_{k-1}\|_{\bar{W}_1}\right)}.$$

*The conclusion still holds for the CTD case if we take $K \geq \frac{4}{\varepsilon^2(1-\gamma)^2} + 1$.*

The proof can be found in Appendix C.3. Combining the two results, we find the following recurrence relation in terms of the $\bar{W}_1$ norm holds given the choice of $T$, with probability at least $1-\delta$, for all $t \geq \frac{T}{c_6 \log T}$

$$\|\Delta_t\|_{\bar{W}_1} \leq \frac{1}{\sqrt{1-\gamma}} \|\Delta_t\| \leq 35 \sqrt{\frac{(\log^3 T) \left(\log \frac{|\mathcal{S}|T}{\delta}\right)}{(1-\gamma)^3 T} \left(1 + \max_{k:\, t/2 < k \leq t} \|\Delta_{k-1}\|_{\bar{W}_1}\right)}.$$

In Theorem C.1, we solve the relation and obtain the error bound of the last iterate estimator:

$$\|\Delta_T\|_{\bar{W}_1} \leq C_7 \left( \sqrt{\frac{(\log^3 T) \left(\log \frac{|\mathcal{S}|T}{\delta}\right)}{(1-\gamma)^3 T}} + \frac{(\log^3 T) \left(\log \frac{|\mathcal{S}|T}{\delta}\right)}{(1-\gamma)^3 T} \right),$$

where $C_7 > 1$ is a large universal constant depending on $c_6$. Now, we can obtain the conclusion if taking $C_4 \geq 2C_7^2$ and $T \geq \frac{C_4 \log^3 T}{\varepsilon^2(1-\gamma)^3} \log \frac{|\mathcal{S}|T}{\delta}$.

## 6 Conclusions

In this paper we have studied the statistical performance of the distributional temporal difference learning (TD) from a non-asymptotic perspective. Specifically, we have considered two instances of distributional TD, namely, the non-parametric distributional TD (NTD) and the categorical distributional TD (CTD). For both NTD and CTD, we have shown that $\widetilde{O}\left(\frac{1}{\varepsilon^{2p}(1-\gamma)^{2p+1}}\right)$ iterations are sufficient to achieve a $p$-Wasserstein $\varepsilon$-optimal estimator, which is minimax optimal (up to logarithmic factors). We have established a novel Freedman's inequality in Hilbert spaces to prove these theoretical results, which has independent theoretical value beyond the current work. We leave the details to Appendix A.

## Acknowledgments and Disclosure of Funding

This work has been supported by the National Key Research and Development Project of China (No. 2022YFA1004002), the National Natural Science Foundation of China (No. 12271011 and No. 12350001), and the MOE Project of Key Research Institute of Humanities and Social Sciences (No.22JJD110001).

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

# A The Key Lemma: Freedman's Inequality in Hilbert Spaces

Freedman's inequality, proposed in [Freedman, 1975], can be viewed as a Bernstein's inequality for martingales, which is crucial for analyzing stochastic approximation algorithms. Compared to the Azuma-Hoeffding inequality which only utilizes the boundedness of martingale difference sequences, Freedman's inequality incorporates second-order information, namely the quadratic variation (cumulative conditional variance) of martingales. This may leads to a sharper concentration result. It has various generalizations, such as matrix Freedman's inequality [Tropp, 2011]. However, to the best of our knowledge, a Freedman's inequality in Hilbert spaces has not been established yet. Just as Freedman's inequality is essential for the theory of TD (Theorem 1 in [Li et al., 2024]), it is indispensable for deriving the minimax non-asymptotic convergence bound for distributional TD.

In this section, we will present a Freedman's inequalities in Hilbert spaces. Firstly, we will state a Hilbert space version of the original Freedman's inequality (Theorem 1.6 in [Freedman, 1975]). After that, we state a generalization of a more powerful version (Theorem 6 in [Li et al., 2024]) to Hilbert spaces. We will provide self-contained proofs in Appendix A.1, primarily inspired by Theorem 3.2 in [Pinelis, 1994]. The necessary knowledge of martingale theory for the proofs can be found in any standard textbook, such as [Durrett, 2019].

Let $\mathcal{X}$ be a Hilbert space, $\{X_i\}_{i=1}^n$ be an $\mathcal{X}$-valued martingale difference sequence adapted to the filtration $\{\mathcal{F}_i\}_{i=1}^n$, $Y_i := \sum_{j=1}^i X_j$ be the corresponding martingale, $W_i := \sum_{j=1}^i \sigma_j^2$ be the corresponding quadratic variation process. Here $\sigma_j^2 := \mathbb{E}_{j-1} \|X_j\|^2$, and $\mathbb{E}_i [\cdot] := \mathbb{E}[\cdot|\mathcal{F}_i]$ is the conditional expectation.

**Theorem A.1** (Freedman's inequality in Hilbert spaces). *Suppose $\max_{i\in[n]} \|X_i\| \le b$ for some constant $b > 0$. Then, for any $\varepsilon, \sigma > 0$, the following inequality holds*

$$\mathbb{P}\left(\exists k \in [n], s.t. \ \|Y_k\| \ge \varepsilon \text{ and } W_k \le \sigma^2\right) \le 2\exp\left\{-\frac{\varepsilon^2/2}{\sigma^2 + b\varepsilon/3}\right\}. \qquad (4)$$

Now, we are ready to state the generalization of Theorem 6 in [Li et al., 2024] to Hilbert spaces, which is used in our non-asymptotic analysis.

**Theorem A.2** (Freedman's inequality in Hilbert spaces with bounded quadratic variation). *Suppose $\max_{i\in[n]} \|X_i\| \le b$ and $W_n \le \sigma^2$ for some constants $b, \sigma > 0$ almost surely. Then, for any $\delta \in (0,1)$, and any positive integer $H \ge 1$, with probability at least $1 - \delta$, for all $k \in [n]$, the following inequality holds*

$$\|Y_k\| \le \sqrt{8 \max\left\{W_k, \frac{\sigma^2}{2H}\right\} \log \frac{2H}{\delta}} + \frac{4}{3} b \log \frac{2H}{\delta}. \qquad (5)$$

The proof can be found in Appendix A.2.

**Remark 2:** Theorem 2.1 can be straightforwardly extended to the case where $(\|X_i\|)_{i=1}^n$ satisfies the Bernstein condition (Theorem 1.2A in [de la Pena, 1999]), thereby relaxing the boundedness assumption on $\|X_i\|$. Namely, $\mathbb{E}_{i-1} \|X_i\|^k \le \frac{1}{2} k! \sigma_i^2 b^{k-2}$ for some $b > 0$, and for all $i \in [n]$, $k \in \{2, 3, \cdots\}$. In this case, Freedman's inequality still holds, albeit with a worse constant.

$$\mathbb{P}\left(\exists k \in [n], \text{s.t. } \|Y_k\| \ge \varepsilon \text{ and } W_k \le \sigma^2\right) \le 2\exp\left\{-\frac{\varepsilon^2/2}{\sigma^2 + b\varepsilon}\right\}. \qquad (6)$$

The proof only requires making appropriate modifications after the fifth line of Equation (12). Note that Bernstein condition holds if $\max_{i\in[n]} \|X_i\| \le b$.

## A.1 Proof of Theorem 2.1

*Proof.* For any $\lambda > 0$, $t \in [0, 1]$ and $j \in [n]$, let $\phi(t) = \phi_{j,\lambda}(t) := \mathbb{E}_{j-1} \cosh(\lambda \|Y_{j-1} + tX_j\|) = \mathbb{E}_{j-1} \cosh(\lambda u(t))$, where $u(t) := \|Y_{j-1} + tX_j\|$. We aim to use the Newton-Leibniz formula to establish the relationship between $\phi(1) = \mathbb{E}_{j-1} \cosh(\lambda \|Y_j\|)$ and $\phi(0) = \cosh(\lambda \|Y_{j-1}\|)$. This will allow us to construct a positive supermartingale $(B_i)_{i=0}^n$. By utilizing the positive supermartingale and optional stopping theorem, we can derive the desired concentration inequality.

Firstly, we calculate the derivative of $\phi$.

$$u'(t) = \frac{\langle Y_{j-1} + tX_j, X_j \rangle}{u(t)}, \tag{7}$$

$$\begin{aligned}
\phi'(t) &= \lambda \mathbb{E}_{j-1} \left[ \sinh \left( \lambda u(t) \right) u'(t) \right] \\
&= \lambda \mathbb{E}_{j-1} \left[ \sinh \left( \lambda u(t) \right) \frac{\langle Y_{j-1} + tX_j, X_j \rangle}{u(t)} \right],
\end{aligned} \tag{8}$$

$$\begin{aligned}
\phi'(0) &= \lambda \mathbb{E}_{j-1} \left[ \sinh \left( \lambda u(0) \right) \frac{\langle Y_{j-1}, X_j \rangle}{u(0)} \right] \\
&= \lambda \sinh \left( \lambda \|Y_{j-1}\| \right) \frac{\langle Y_{j-1}, \mathbb{E}_{j-1} [X_j] \rangle}{\|Y_{j-1}\|} \\
&= 0.
\end{aligned} \tag{9}$$

By utilizing Newton-Leibniz formula, we have

$$\begin{aligned}
\phi(1) &= \phi(0) + \int_0^1 \phi'(s) ds \\
&= \phi(0) + \int_0^1 \int_0^s \phi''(t) dt \, ds \\
&= \phi(0) + \int_0^1 (1-t) \phi''(t) dt.
\end{aligned} \tag{10}$$

Now, we calculate the second order derivate of $\phi$.

$$\begin{aligned}
\phi''(t) &= \lambda \mathbb{E}_{j-1} \left\{ \frac{d}{dt} \left[ \sinh \left( \lambda u(t) \right) u'(t) \right] \right\} \\
&= \lambda \mathbb{E}_{j-1} \left[ \lambda \left( u'(t) \right)^2 \cosh \left( \lambda u(t) \right) + u''(t) \sinh \left( \lambda u(t) \right) \right] \\
&\leq \lambda^2 \mathbb{E}_{j-1} \left[ \left( \left( u'(t) \right)^2 + u''(t) u(t) \right) \cosh \left( \lambda u(t) \right) \right] \\
&= \frac{\lambda^2}{2} \mathbb{E}_{j-1} \left[ \left( u^2 \right)'' (t) \cosh \left( \lambda u(t) \right) \right] \\
&= \lambda^2 \mathbb{E}_{j-1} \left[ \|X_j\|^2 \cosh \left( \lambda \|Y_{j-1} + tX_j\| \right) \right] \\
&\leq \lambda^2 \cosh \left( \lambda \|Y_{j-1}\| \right) \mathbb{E}_{j-1} \left[ \|X_j\|^2 \exp \left( \lambda t \|X_j\| \right) \right],
\end{aligned} \tag{11}$$

where in the third line, we used $u''(t) = \frac{\|X_j\|^2 u(t) - \frac{\langle Y_{j-1} + tX_j, X_j \rangle^2}{u(t)}}{u^2(t)} \geq 0$ by Cauchy-Schwarz inequality, and $h(x) = x \cosh(x) - \sinh(x) \geq 0$ for any $x \geq 0$, the inequality holds because $h(0) = 0$ and $h'(x) = x \sinh(x) \geq 0$ for any $x \geq 0$. In the fourth line, we used $\left( u^2 \right)'' (t) = 2 \left( \left( u'(t) \right)^2 + u''(t) u(t) \right)$. In the fifth line, we used

$$\left( u^2 \right)'' (t) = \frac{d^2}{dt^2} \|Y_{j-1} + tX_j\|^2 = \frac{d}{dt} \left( 2 \langle Y_{j-1} + tX_j, X_j \rangle \right) = 2 \|X_j\|^2.$$

And in the last line, we used

$$\cosh \left( \lambda \|Y_{j-1} + tX_j\| \right) \leq \cosh \left( \lambda \|Y_{j-1}\| \right) \exp \left( \lambda t \|X_j\| \right),$$

this holds since

$$\exp \left( \lambda \|Y_{j-1} + tX_j\| \right) \leq \exp \left\{ \lambda \left( \|Y_{j-1}\| + t \|X_j\| \right) \right\} = \exp \left( \lambda \|Y_{j-1}\| \right) \exp \left( \lambda t \|X_j\| \right),$$
$$\exp \left( -\lambda \|Y_{j-1} + tX_j\| \right) \leq \exp \left\{ -\lambda \left( \|Y_{j-1}\| - t \|-X_j\| \right) \right\} = \exp \left( -\lambda \|Y_{j-1}\| \right) \exp \left( \lambda t \|X_j\| \right).$$

Hence, we can derive the following inequality for all $j \in [n]$

$$
\begin{aligned}
\mathbb{E}_{j-1}\left[\cosh\left(\lambda\|Y_j\|\right)\right] &= \phi(1) = \phi(0) + \int_0^1 (1-t)\phi''(t)dt \\
&\leq \cosh\left(\lambda\|Y_{j-1}\|\right) + \lambda^2 \cosh\left(\lambda\|Y_{j-1}\|\right)\mathbb{E}_{j-1}\left[\|X_j\|^2 \int_0^1 (1-t)\exp\left(\lambda t\|X_j\|\right)dt\right] \\
&= \cosh\left(\lambda\|Y_{j-1}\|\right) + \lambda^2 \cosh\left(\lambda\|Y_{j-1}\|\right)\mathbb{E}_{j-1}\left[\|X_j\|^2 \frac{\exp\left(\lambda\|X_j\|\right) - \lambda\|X_j\| - 1}{\lambda^2\|X_j\|^2}\right] \\
&= \mathbb{E}_{j-1}\left[\exp\left(\lambda\|X_j\|\right) - \lambda\|X_j\|\right]\cosh\left(\lambda\|Y_{j-1}\|\right) \\
&= \mathbb{E}_{j-1}\left[1 + \sum_{k=0}^{\infty}\frac{1}{(k+2)!}\left(\lambda\|X_j\|\right)^{k+2}\right]\cosh\left(\lambda\|Y_{j-1}\|\right) \\
&\leq \mathbb{E}_{j-1}\left[1 + \frac{\lambda^2\|X_j\|^2}{2}\sum_{k=0}^{\infty}\left(\frac{\lambda b}{3}\right)^k\right]\cosh\left(\lambda\|Y_{j-1}\|\right) \\
&= \left(1 + \frac{\lambda^2\sigma_j^2}{2(1-\lambda b/3)}\right)\cosh\left(\lambda\|Y_{j-1}\|\right) \\
&\leq \exp\left\{\frac{\lambda^2\sigma_j^2}{2(1-\lambda b/3)}\right\}\cosh\left(\lambda\|Y_{j-1}\|\right),
\end{aligned}
\tag{12}
$$

which holds for any $\lambda \in (0, \frac{3}{b})$. In the fifth line, we used Taylor expansion $e^x = \sum_{k=0}^{\infty}\frac{x^k}{k!}$. In the sixth line, we used $(k+2)! \geq 2(3^k)$ and $\|X_j\| \leq b$. In the seventh line, we used Taylor expansion $\frac{1}{1-x} = \sum_{k=0}^{\infty} x^k$ for $x \in (-1, 1)$.

Let $B_0 := 1$, $B_i := \exp\left\{-\frac{\lambda^2 W_i}{2(1-\lambda b/3)}\right\}\cosh\left(\lambda\|Y_i\|\right)$, then

$$
\begin{aligned}
\mathbb{E}_{i-1}\left[B_i\right] &= \exp\left\{-\frac{\lambda^2 W_{i-1}}{2(1-\lambda b/3)}\right\}\exp\left\{-\frac{\lambda^2\sigma_i^2}{2(1-\lambda b/3)}\right\}\mathbb{E}_{i-1}\left[\cosh\left(\lambda\|Y_i\|\right)\right] \\
&\leq \exp\left\{-\frac{\lambda^2 W_{i-1}}{2(1-\lambda b/3)}\right\}\cosh\left(\lambda\|Y_{i-1}\|\right) \\
&= B_{i-1},
\end{aligned}
\tag{13}
$$

i.e., $(B_i)_{i=0}^n$ is positive supermartingale. By optional stopping theorem (Theorem 4.8.4 in [Durrett, 2019]), for any stopping time $\tau$, we have $\mathbb{E}\left[B_\tau\right] \leq \mathbb{E}\left[B_0\right] = 1$.

Let $\tau := \inf\left\{k \in [n] : \|Y_k\| \geq \varepsilon\right\}$ be a stopping time, and $\inf\emptyset := \infty$. Define an event

$$
A := \left\{\exists k \in [n], \text{s.t. } \|Y_k\| \geq \varepsilon \text{ and } W_k \leq \sigma^2\right\},
\tag{14}
$$

then on $A$, we have $\tau < \infty$, $\|Y_\tau\| \geq \varepsilon$ and $W_\tau \leq \sigma^2$, noting that $W_k$ is non-decreasing with $k$. Our goal is to provide an upper bound for $\mathbb{P}(A)$.

$$
\begin{aligned}
\mathbb{P}(A) &= \mathbb{E}\left[\sqrt{B_\tau}\frac{1}{\sqrt{B_\tau}}\mathbb{1}(A)\right] \\
&\leq \sqrt{\mathbb{E}\left[B_\tau\right]\mathbb{E}\left[\frac{1}{B_\tau}\mathbb{1}(A)\right]} \\
&\leq \sqrt{\mathbb{E}\left[\frac{\exp\left\{\frac{\lambda^2 W_\tau}{2(1-\lambda b/3)}\right\}}{\cosh\left(\lambda\|Y_\tau\|\right)}\mathbb{1}(A)\right]} \\
&\leq \sqrt{\mathbb{E}\left[\frac{\exp\left\{\frac{\lambda^2 \sigma^2}{2(1-\lambda b/3)}\right\}}{\cosh\left(\lambda\varepsilon\right)}\mathbb{1}(A)\right]} \\
&\leq \sqrt{2\exp\left\{-\lambda\varepsilon + \frac{\lambda^2\sigma^2}{2(1-\lambda b/3)}\right\}\mathbb{P}(A)},
\end{aligned}
\tag{15}
$$

where in the second line, we used Cauchy-Schwarz inequality. In the third line, we used $\mathbb{E}\left[B_\tau\right] \leq 1$. In the fourth line, we used $\|Y_\tau\| \geq \varepsilon$ and $W_\tau \leq \sigma^2$ on $A$, and $\cosh(x)$ is increasing when $x \geq 0$. In the last line, we used $\cosh(x) \geq \frac{1}{2}e^x$.

Hence for any $\lambda \in (0, \frac{3}{b})$

$$
\mathbb{P}(A) \leq 2\exp\left\{-\lambda\varepsilon + \frac{\lambda^2\sigma^2}{2\left(1-\lambda b/3\right)}\right\},
\tag{16}
$$

we can choose $\lambda^\star = \frac{\varepsilon}{\sigma^2 + \varepsilon b/3} \in (0, \frac{3}{b})$, then

$$
\begin{aligned}
\mathbb{P}(A) &\leq 2\exp\left\{-\lambda^\star\varepsilon + \frac{\left(\lambda^\star\right)^2 \sigma^2}{2\left(1-\lambda^\star b/3\right)}\right\} \\
&= 2\exp\left\{-\frac{\varepsilon^2}{\sigma^2 + \varepsilon b/3} + \frac{\sigma^2}{2\left(1 - \frac{\varepsilon b/3}{\sigma^2 + \varepsilon b/3}\right)}\frac{\varepsilon^2}{\left(\sigma^2 + \varepsilon b/3\right)^2}\right\} \\
&= 2\exp\left\{-\frac{\varepsilon^2/2}{\sigma^2 + \varepsilon b/3}\right\},
\end{aligned}
\tag{17}
$$

which is the desired conclusion. $\qquad\square$

## A.2 Proof of Theorem A.2

*Proof.* According to Theorem 2.1, for any $\varepsilon, \tilde{\sigma} > 0$, we have

$$
\mathbb{P}\left(\exists k \in [n],\ \|Y_k\| \geq \varepsilon \text{ and } W_k \leq \tilde{\sigma}^2\right) \leq 2\exp\left\{-\frac{\varepsilon^2/2}{\tilde{\sigma}^2 + b\varepsilon/3}\right\}.
\tag{18}
$$

We can check that when $\varepsilon = \sqrt{4\tilde{\sigma}^2 \log\frac{2}{\delta}} + \frac{4}{3}b\log\frac{2}{\delta}$, the upper bound on RHS is less than $\delta$. Hence,

$$
\mathbb{P}\left(\exists k \in [n],\ \|Y_k\| \geq \sqrt{4\tilde{\sigma}^2 \log\frac{2}{\delta}} + \frac{4}{3}b\log\frac{2}{\delta} \text{ and } W_k \leq \tilde{\sigma}^2\right) \leq \delta.
\tag{19}
$$

For each $k \in [n]$, define the events

$$\mathcal{H}_H^{(k)} := \left\{ \|Y_k\| \geq \sqrt{8 \max\left\{ W_k, \frac{\sigma^2}{2^H} \right\} \log \frac{2H}{\delta}} + \frac{4}{3} b \log \frac{2H}{\delta} \right\},$$

$$\mathcal{B}_{H,H}^{(k)} := \left\{ \|Y_k\| \geq \sqrt{4 \frac{\sigma^2}{2^{H-1}} \log \frac{2H}{\delta}} + \frac{4}{3} b \log \frac{2H}{\delta} \text{ and } W_k \leq \frac{\sigma^2}{2^{H-1}} \right\},$$

$$\mathcal{B}_{h,H}^{(k)} := \left\{ \|Y_k\| \geq \sqrt{4 \frac{\sigma^2}{2^{h-1}} \log \frac{2H}{\delta}} + \frac{4}{3} b \log \frac{2H}{\delta} \text{ and } \frac{\sigma^2}{2^h} \leq W_k \leq \frac{\sigma^2}{2^{h-1}} \right\}, \quad 1 \leq h \leq H - 1. \tag{20}$$

By the definition, we only need to show $\mathbb{P}\left( \bigcup_{k \in [n]} \mathcal{H}_H^{(k)} \right) \leq \delta$. Since $W_k \leq W_n \leq \sigma^2$ almost surely, we can find that $\mathcal{H}_H^{(k)} \subseteq \bigcup_{h \in [H]} \mathcal{B}_{h,H}^{(k)}$ (we will justify this later). Then $\bigcup_{k \in [n]} \mathcal{H}_H^{(k)} \subseteq \bigcup_{h \in [H]} \bigcup_{k \in [n]} \mathcal{B}_{h,H}^{(k)}$. By the inequality (19) with $\tilde{\sigma}^2 = \frac{\sigma^2}{2^{h-1}}$ and setting $\delta$ as $\frac{\delta}{H}$, we have $\mathbb{P}\left( \bigcup_{k \in [n]} \mathcal{B}_{h,H}^{(k)} \right) \leq \frac{\delta}{H}$ for all $h \in [H]$. By the union bound, we can arrive at the conclusion:

$$\mathbb{P}\left( \bigcup_{k \in [n]} \mathcal{H}_H^{(k)} \right) \leq \sum_{h=1}^{H} \mathbb{P}\left( \bigcup_{k \in [n]} \mathcal{B}_{h,H}^{(k)} \right) \leq \delta. \tag{21}$$

To justify $\mathcal{H}_H^{(k)} \subseteq \bigcup_{h \in [H]} \mathcal{B}_{h,H}^{(k)}$, we can consider the decomposition

$$\mathcal{H}_H^{(k)} = \bigcup_{h \in [H]} \left( \mathcal{H}_H^{(k)} \cap \mathcal{C}_{h,H}^{(k)} \right), \tag{22}$$

where

$$\mathcal{C}_{H,H}^{(k)} := \left\{ W_k \leq \frac{\sigma^2}{2^{H-1}} \right\}, \quad \mathcal{C}_{h,H}^{(k)} := \left\{ \frac{\sigma^2}{2^h} \leq W_k \leq \frac{\sigma^2}{2^{h-1}} \right\}, \quad 1 \leq h \leq H - 1. \tag{23}$$

The decomposition holds because $W_k \leq W_n \leq \sigma^2$ almost surely. We only need to show that for each $h \in [H]$,

$$\mathcal{H}_H^{(k)} \cap \mathcal{C}_{h,H}^{(k)} \subseteq \mathcal{B}_{h,H}^{(k)}. \tag{24}$$

On the event $\mathcal{H}_H^{(k)} \cap \mathcal{C}_{h,H}^{(k)}$, we have

$$\|Y_k\| \geq \sqrt{8 \max\left\{ W_k, \frac{\sigma^2}{2^H} \right\} \log \frac{2H}{\delta}} + \frac{4}{3} b \log \frac{2H}{\delta}$$
$$\geq \sqrt{4 \frac{\sigma^2}{2^{h-1}} \log \frac{2H}{\delta}} + \frac{4}{3} b \log \frac{2H}{\delta}, \tag{25}$$

hence $\mathcal{H}_H^{(k)} \cap \mathcal{C}_{h,H}^{(k)} \subseteq \mathcal{B}_{h,H}^{(k)}$. $\qquad\square$

## B  Minimax Lower Bound of Distributional Policy Evaluation

In this section, we still consider infinite-horizon tabular MDP defined in Section 2, and assume a generative model is accessible. For any positive integer $D$, we define $\mathfrak{M}(D)$ as the set of all MDPs with state space size $|\mathcal{S}| = D$. For any MDP $M$ and policy $\pi$, we denote $V_M^\pi$ as the corresponding value function, and $\eta_M^\pi$ as the corresponding return distribution.

Now, we can state the minimax lower bound of the distributional policy evaluation task in the 1-Wasserstein metric.

**Theorem B.1** (Minimax lower bound of distributional policy evaluation in the 1-Wasserstein metric)**.** *For any positive integer $D \geq 3$, and sample size $T \geq \frac{C}{1-\gamma} \log \frac{D}{2}$, the following result holds*

$$\inf_{\hat{\eta}} \sup_{M \in \mathfrak{M}(D)} \sup_{\pi} \mathbb{E}\left[ \bar{W}_1\left( \hat{\eta}, \eta_M^\pi \right) \right] \geq \frac{c}{(1-\gamma)^{3/2}} \sqrt{\frac{\log \frac{D}{2}}{T}}.$$

Here, $c, C > 0$ are universal constants, and the infimum $\hat{\eta} \in \mathscr{P}^D$ ranges over all measurable functions of $T$ samples from the generative model.

The theorem states that for any algorithm, there exist corresponding MDP $M$ and policy $\pi$, such that to ensure $\mathbb{E}\left[\bar{W}_1\left(\hat{\eta}, \eta_M^\pi\right)\right] \leq \varepsilon$ for some $\varepsilon > 0$, at least $\tilde{\Omega}\left(\frac{1}{\varepsilon^2(1-\gamma)^3}\right)$ samples are required.

*Proof of Theorem B.1.* For any $\eta \in \mathscr{P}^D$, we define $\mathcal{V}(\eta) \in \mathbb{R}^D$ as the entry-wise expectation of $\eta$. It is easy to check that $\mathcal{V}(\eta_M^\pi) = V_M^\pi$. And recall the dual representation of 1-Wasserstein metric (Corollary 5.16 in [Villani et al., 2009])

$$W_1(\mu, \nu) = \sup_{f \,:\, f \text{ is 1-Lipschitz}} \left|\mathbb{E}_{X \sim \mu}[f(X)] - \mathbb{E}_{Y \sim \nu}[f(Y)]\right|, \quad \forall \mu, \nu \in \mathscr{P}, \tag{26}$$

we have $\bar{W}_1(\hat{\eta}, \eta_M^\pi) \geq \|\mathcal{V}(\hat{\eta}) - V_M^\pi\|_\infty$. Hence

$$\inf_{\hat{\eta}} \sup_{M \in \mathfrak{M}(D)} \sup_\pi \mathbb{E}\left[\bar{W}_1\left(\hat{\eta}, \eta_M^\pi\right)\right] \geq \inf_{\hat{\eta}} \sup_{M \in \mathfrak{M}(D)} \sup_\pi \mathbb{E}\left[\|\mathcal{V}(\hat{\eta}) - V_M^\pi\|_\infty\right]$$

$$\geq \inf_{\hat{V}} \sup_{M \in \mathfrak{M}(D)} \sup_\pi \mathbb{E}\left[\left\|\hat{V} - V_M^\pi\right\|_\infty\right] \tag{27}$$

$$\geq \frac{c}{(1-\gamma)^{3/2}} \sqrt{\frac{\log \frac{D}{2}}{T}},$$

where the second inequality holds because $\mathcal{V}(\hat{\eta}) \in \mathbb{R}^D$ is also a measurable function of $T$ samples from the generative model, and the infimum $\hat{V} \in \mathbb{R}^D$ ranges over all measurable functions of $T$ samples from the generative model. And the last inequality is due to Theorem 2(b) in [Pananjady and Wainwright, 2020]. $\qquad\square$

## C  Omitted Proofs in Section 5

### C.1  Remove the Dependence on T for Step Sizes

We have shown that the conclusion holds for

$$T \geq \frac{C_4 \log^3 T}{\varepsilon^2(1-\gamma)^3} \log \frac{|\mathcal{S}| T}{\delta}, \tag{28}$$

$$\frac{1}{1 + \frac{c_5(1-\sqrt{\gamma})T}{\log^2 T}} \leq \alpha_t \leq \frac{1}{1 + \frac{c_6(1-\sqrt{\gamma})t}{\log^2 T}}, \tag{29}$$

where $c_5 c_6 \leq \frac{1}{8}$, $c_5 > c_6 > 0$ and $C_4 > 0$.

Then for some $c_2 > c_3 > 0$ to be determined, now we assume

$$\frac{1}{1 + \frac{c_2(1-\sqrt{\gamma})t}{\log^2 t}} \leq \alpha_t \leq \frac{1}{1 + \frac{c_3(1-\sqrt{\gamma})t}{\log^2 t}}. \tag{30}$$

Next, we will show that if we consider the result of the $\frac{T}{2}$-th iteration with this step size scheme as the initialization of a new iteration process, then the step sizes in the subsequent $\frac{T}{2}$ iterations lie in the previously established range. If this is done, the conclusion still holds if we choose $T \geq \frac{2C_4 \log^3 T}{\varepsilon^2(1-\gamma)^3} \log \frac{|\mathcal{S}| T}{\delta}$, since the initialization $\eta_{T/2}^\pi \in \mathscr{P}^{\mathcal{S}}$ (or $\mathscr{P}_K^{\mathcal{S}}$ in the case of CTD) is independent of the samples obtained for $\frac{T}{2} < t \leq T$.

For any $\frac{T}{2} < t \leq T$, we denote $\tau := t - \frac{T}{2}$, we can see that there exist $c_2 > c_3 > 0$, such that the last inequality in both of the following lines hold simultaneously, which is desired.

$$\tilde{\alpha}_\tau := \alpha_t \leq \frac{1}{1 + \frac{c_3(1-\sqrt{\gamma})(\tau+T/2)}{\log^2(\tau+T/2)}} \leq \frac{1}{1 + \frac{c_3(1-\sqrt{\gamma})\tau}{\log^2 T}} \leq \frac{1}{1 + \frac{c_6(1-\sqrt{\gamma})\tau}{\log^2(T/2)}}, \tag{31}$$

and

$$\tilde{\alpha}_\tau = \alpha_t \geq \frac{1}{1 + \frac{c_2(1-\sqrt{\gamma})(\tau+T/2)}{\log^2(\tau+T/2)}} \geq \frac{1}{1 + \frac{2c_2(1-\sqrt{\gamma})T/2}{\log^2(T/2)}} \geq \frac{1}{1 + \frac{c_5(1-\sqrt{\gamma})T/2}{\log^2(T/2)}}. \tag{32}$$

## C.2 Range of Step Size

*Proof of Lemma 5.1.*

$$(1 - \sqrt{\gamma})\alpha_t \geq \frac{1 - \sqrt{\gamma}}{1 + \frac{c_5(1 - \sqrt{\gamma})T}{\log^2 T}} \geq \frac{1 - \sqrt{\gamma}}{\frac{2c_5(1 - \sqrt{\gamma})T}{\log^2 T}} = \frac{\log^2 T}{2c_5 T}. \tag{33}$$

For any $0 \leq k \leq \frac{t}{2}$,

$$\begin{aligned}
\beta_k^{(t)} &\leq \left[1 - \alpha_{t/2}(1 - \sqrt{\gamma})\right]^{t/2} \\
&\leq \left(1 - \frac{\log^2 T}{2c_5 T}\right)^{t/2} \\
&\leq \left(1 - \frac{\log^2 T}{2c_5 T}\right)^{\frac{T}{2c_6 \log T}} \\
&= \left\{\left(1 - \frac{\log^2 T}{2c_5 T}\right)^{\frac{2c_5 T}{\log^2 T}}\right\}^{\frac{\log T}{4c_5 c_6}} \\
&\leq \frac{1}{T^2},
\end{aligned} \tag{34}$$

where in the last inequality, we used $c_5 c_6 \leq \frac{1}{8}$.

And for any $\frac{t}{2} < k \leq t$,

$$\beta_k^{(t)} \leq \alpha_k \leq \frac{1}{\frac{c_6(1 - \sqrt{\gamma})k}{\log^2 T}} \leq \frac{2\log^3 T}{(1 - \sqrt{\gamma})T}. \tag{35}$$

$\square$

## C.3 Concentration of the Martingale Term

*Proof of Lemma 5.2.* We will show that the inequality holds for each $t \geq \frac{T}{c_6 \log T}$ and then apply the union bound. For any $s \in \mathcal{S}$, we denote

$$\zeta_k(s) := \zeta_k^{(t)}(s) = \alpha_k \left\{ \prod_{i=k+1}^{t} \left[(1 - \alpha_i)\mathcal{I} + \alpha_i \mathcal{T}\right] (\mathcal{T}_k - \mathcal{T})\eta_{k-1} \right\}(s), \tag{36}$$

where we omit the superscript $(t)$ for brevity, then LHS in the lemma equals $\left\|\sum_{k=1}^{t} \zeta_k\right\|$ for each $t$. Let $\mathcal{F}_k$ denote the $\sigma$-field that contains all information up to time step $k$, then $\{\zeta_k(s)\}_{k=1}^{t}$ is a $\{\mathcal{F}_k\}_{k=1}^{t}$-martingale difference sequence:

$$\mathbb{E}_{k-1}\left[\zeta_k(s)\right] = \alpha_k \left\{ \prod_{i=k+1}^{t} \left[(1 - \alpha_i)\mathcal{I} + \alpha_i \mathcal{T}\right] \mathbb{E}_{k-1}\left[(\mathcal{T}_k - \mathcal{T})\eta_{k-1}\right] \right\}(s) = 0. \tag{37}$$

the first equality holds because a Bochner integral can be exchanged with a bounded linear operator (see Pisier [2016] for more details about Bochner integral), and the second equality holds due to the definition of the empirical distributional Bellman operator.

We hope to use Freedman's inequality (Theorem A.2) to bound this martingale. To this end, we need to give a deterministic upper bound of the martingale difference sequence, and an upper bound of its quadratic variation.

**Deterministic upper bound of** $\max_{k\in[t]}\|\zeta_k(s)\|$**.** The norm of the martingale difference $\|\zeta_k(s)\|$ can be bounded as follow

$$
\begin{aligned}
\|\zeta_k(s)\| &\leq \|\zeta_k\| \\
&\leq \alpha_k \left\| \prod_{i=k+1}^{t} [(1-\alpha_i)\mathcal{I} + \alpha_i\mathcal{T}] \right\| \|(\mathcal{T}_k - \mathcal{T})\eta_{k-1}\| \\
&\leq \alpha_k \prod_{i=k+1}^{t} ((1-\alpha_i) + \alpha_i\sqrt{\gamma}) \frac{1}{\sqrt{1-\gamma}} \\
&= \frac{\beta_k^{(t)}}{\sqrt{1-\gamma}}.
\end{aligned}
\tag{38}
$$

Hence, $\max_{k\in[t]}\|\zeta_k(s)\| \leq \frac{\max_{k\in[t]}\beta_k^{(t)}}{\sqrt{1-\gamma}} \leq \frac{1}{\sqrt{1-\gamma}}\max\left\{\frac{1}{T^2}, \frac{2\log^3 T}{(1-\sqrt{\gamma})T}\right\} \leq \frac{4\log^3 T}{(1-\gamma)^{3/2}T} =: b.$

**Upper bound of quadratic variation.** Now, let's calculate the quadratic variation.

We first introduce some notations. For any $k \in \mathbb{N}$, we denote $\mathsf{Var}(\boldsymbol{\xi}) := \left(\mathbb{E}\left[\|\boldsymbol{\xi}(s)\|^2\right]\right)_{s\in\mathcal{S}} \in \mathbb{R}^{\mathcal{S}}$, $\mathsf{Var}_k(\boldsymbol{\xi}) := \left(\mathbb{E}_k\left[\|\boldsymbol{\xi}(s)\|^2\right]\right)_{s\in\mathcal{S}} \in \mathbb{R}^{\mathcal{S}}$ for any random element $\boldsymbol{\xi}$ in $\mathcal{M}^{\mathcal{S}}$.

For any $\xi \in \mathcal{M}^{\mathcal{S}}$, we define its one-step update Cramér variation as $\boldsymbol{\sigma}(\xi) := \mathsf{Var}\left((\widehat{\mathcal{T}} - \mathcal{T})\xi\right) \in \mathbb{R}^{\mathcal{S}}$, where $\widehat{\mathcal{T}}$ is a random operator and has the same distribution as $\mathcal{T}_1$.

For any $\boldsymbol{x}, \boldsymbol{y} \in \mathbb{R}^{\mathcal{S}}$, we say $\boldsymbol{x} \leq \boldsymbol{y}$ if $\boldsymbol{x}(s) \leq \boldsymbol{y}(s)$ for all $s \in \mathcal{S}$. In this part, $\|\boldsymbol{x}\| := \|\boldsymbol{x}\|_{\infty} = \max_{s\in\mathcal{S}}|\boldsymbol{x}(s)|$, $\sqrt{\boldsymbol{x}} := \left(\sqrt{\boldsymbol{x}(s)}\right)_{s\in\mathcal{S}}$. And for any $\boldsymbol{U} \in \mathbb{R}^{\mathcal{S}\times\mathcal{S}}$, $\|\boldsymbol{U}\| := \|\boldsymbol{U}\|_{\infty} = \sup_{\boldsymbol{x}\in\mathbb{R}^{\mathcal{S}}, \|\boldsymbol{x}\|=1}\|\boldsymbol{U}\boldsymbol{x}\| = \max_{s\in\mathcal{S}}\sum_{s'\in\mathcal{S}}|\boldsymbol{U}(s,s')|$.

For any $\{\boldsymbol{x}_k\}_{k=1}^{n} \subset \mathbb{R}^{\mathcal{S}}$, we denote $\max_{k\in[n]}\boldsymbol{x}_k$ as $\left(\max_{k\in[n]}\boldsymbol{x}_k(s)\right)_{s\in\mathcal{S}}$.

We denote $\boldsymbol{I} \in \mathbb{R}^{\mathcal{S}\times\mathcal{S}}$ as the identity matrix, $\boldsymbol{1} \in \mathbb{R}^{\mathcal{S}}$ as the all-ones vector, and $\boldsymbol{P} := P^{\pi} \in \mathbb{R}^{\mathcal{S}\times\mathcal{S}}$, i.e., $\boldsymbol{P}(s, s') := P^{\pi}(s'|s) = \sum_{a\in\mathcal{A}}\pi(a|s)P(s'|s, a)$.

With these notations, the quadratic variation is $\boldsymbol{W}_t := \sum_{k=1}^{t}\mathsf{Var}_{k-1}(\zeta_k)$. To bound the quadratic variation $\boldsymbol{W}_t$, we need to bound $\mathsf{Var}_{k-1}(\zeta_k)$.

**Lemma C.1.**

$$
\mathsf{Var}_{k-1}(\zeta_k) \leq \alpha_k \beta_k^{(t)} \prod_{i=k+1}^{t} [(1-\alpha_i)\boldsymbol{I} + \alpha_i\sqrt{\gamma}\boldsymbol{P}] \boldsymbol{\sigma}(\eta_{k-1}).
$$

Hence, the quadratic variation $\boldsymbol{W}_t$ can be bounded as follow

$$
\begin{aligned}
\boldsymbol{W}_t &= \sum_{k=1}^{t} \mathsf{Var}_{t-1}\left(\zeta_k\right) \\
&\leq \sum_{k=1}^{t} \alpha_k \beta_k^{(t)} \prod_{i=k+1}^{t} \left[(1-\alpha_i)\boldsymbol{I} + \alpha_i\sqrt{\gamma}\boldsymbol{P}\right]\boldsymbol{\sigma}(\eta_{k-1}) \\
&\leq \sum_{k=1}^{t/2} \alpha_k \beta_k^{(t)} \left\| \prod_{i=k+1}^{t} \left[(1-\alpha_i)\boldsymbol{I} + \alpha_i\sqrt{\gamma}\boldsymbol{P}\right]\right\| \|\boldsymbol{\sigma}(\eta_{k-1})\| \mathbf{1} + \sum_{k=t/2+1}^{t} \alpha_k \beta_k^{(t)} \prod_{i=k+1}^{t} \left[(1-\alpha_i)\boldsymbol{I} + \alpha_i\sqrt{\gamma}\boldsymbol{P}\right]\boldsymbol{\sigma}(\eta_{k-1}) \\
&\leq \sum_{k=1}^{t/2} \left(\beta_k^{(t)}\right)^2 \frac{1}{1-\gamma}\mathbf{1} + \left(\max_{k:\,t/2<k\leq t}\beta_k^{(t)}\right) \sum_{k=t/2+1}^{t} \alpha_k \prod_{i=k+1}^{t} \left[(1-\alpha_i)\boldsymbol{I} + \alpha_i\sqrt{\gamma}\boldsymbol{P}\right]\boldsymbol{\sigma}(\eta_{k-1}) \\
&\leq \frac{1}{2(1-\gamma)T^3}\mathbf{1} + \frac{2\log^3 T}{(1-\sqrt{\gamma})T}\left\{\sum_{k=t/2+1}^{t} \alpha_k \prod_{i=k+1}^{t} \left[(1-\alpha_i)\boldsymbol{I} + \alpha_i\sqrt{\gamma}\boldsymbol{P}\right]\right\} \max_{k:\,t/2<k\leq t}\boldsymbol{\sigma}(\eta_{k-1}) \\
&\leq \frac{1}{2(1-\gamma)T^3}\mathbf{1} + \frac{4\log^3 T}{(1-\gamma)T}(\boldsymbol{I}-\sqrt{\gamma}\boldsymbol{P})^{-1} \max_{k:\,t/2<k\leq t}\boldsymbol{\sigma}(\eta_{k-1}),
\end{aligned}
\tag{39}
$$

where in the fourth line, we used

$$
\alpha_k \left\|\prod_{i=k+1}^{t}\left[(1-\alpha_i)\boldsymbol{I} + \alpha_i\sqrt{\gamma}\boldsymbol{P}\right]\right\| \leq \alpha_k \prod_{i=k+1}^{t}\left[(1-\alpha_i) + \alpha_i\sqrt{\gamma}\right] = \beta_k^{(t)},
$$

and

$$
\|\boldsymbol{\sigma}(\eta_{k-1})\| \leq \int_0^{\frac{1}{1-\gamma}} dx = \frac{1}{1-\gamma}.
$$

In the last line, we used the fact that $\max_{k:\,t/2\leq k<t}\boldsymbol{\sigma}(\eta_{k-1}) \geq \boldsymbol{0}$ and the following lemma:

**Lemma C.2.** *For any $t \in \mathbb{N}$, $(\alpha_i)_{i\in[t]} \in [0,1]^t$, the following inequality holds entry-wise:*

$$
\sum_{k=t/2+1}^{t} \alpha_k \prod_{i=k+1}^{t} \left[\boldsymbol{I} - \alpha_i\left(\boldsymbol{I} - \sqrt{\gamma}\boldsymbol{P}\right)\right] \leq \left(\boldsymbol{I} - \sqrt{\gamma}\boldsymbol{P}\right)^{-1}.
\tag{40}
$$

According to (39), we have the following deterministic upper bound for $\|\boldsymbol{W}_t\| = \max_{s\in\mathcal{S}}\boldsymbol{W}_t(s)$,

$$
\begin{aligned}
\|\boldsymbol{W}_t\| &\leq \frac{1}{2(1-\gamma)T^3} + \frac{4\log^3 T}{(1-\gamma)T}\left\|(\boldsymbol{I}-\sqrt{\gamma}\boldsymbol{P})^{-1}\right\| \max_{k:\,t/2<k<\leq t}\|\boldsymbol{\sigma}(\eta_{k-1})\| \\
&\leq \frac{1}{2(1-\gamma)T^3} + \frac{8\log^3 T}{(1-\gamma)^3 T} \\
&\leq \frac{9\log^3 T}{(1-\gamma)^3 T} \\
&=: \sigma^2.
\end{aligned}
\tag{41}
$$

Let $H = \left\lceil 2\log_2 \frac{1}{1-\gamma}\right\rceil$, we have

$$
\frac{\sigma^2}{2^H} \leq \frac{9\log^3 T}{(1-\gamma)T}.
\tag{42}
$$

By applying Freedman's inequality (Theorem A.2) and utilizing the union bound over $s \in \mathcal{S}$, we obtain with probability at least $1 - \delta$, for all $t \in [T]$ and $s \in \mathcal{S}$

$$\left( \left\| \sum_{k=1}^{t} \zeta_k(s) \right\| \right)_{s \in \mathcal{S}}$$

$$\leq \sqrt{8 \left( \boldsymbol{W}_t + \frac{\sigma^2}{2^H} \mathbf{1} \right) \log \frac{8|\mathcal{S}|T \log \frac{1}{1-\gamma}}{\delta}} + \frac{4}{3} b \log \frac{8|\mathcal{S}|T \log \frac{1}{1-\gamma}}{\delta} \mathbf{1}$$

$$\leq \sqrt{16 \left( \boldsymbol{W}_t + \frac{9 \log^3 T}{(1-\gamma)T} \mathbf{1} \right) \log \frac{|\mathcal{S}|T}{\delta}} + 3b \log \frac{|\mathcal{S}|T}{\delta} \mathbf{1}$$

$$\leq 8 \sqrt{\frac{\left( \log^3 T \right) \left( \log \frac{|\mathcal{S}|T}{\delta} \right)}{(1-\gamma)T} \left[ (\boldsymbol{I} - \sqrt{\gamma}\boldsymbol{P})^{-1} \max_{k: \, t/2 < k \leq t} \boldsymbol{\sigma}(\eta_{k-1}) + 3 \cdot \mathbf{1} \right]} + \frac{12 \left( \log^3 T \right) \left( \log \frac{|\mathcal{S}|T}{\delta} \right)}{(1-\gamma)^{3/2}T} \mathbf{1},$$

$$(43)$$

where we used $\log \frac{8|\mathcal{S}|T \log \frac{1}{1-\gamma}}{\delta} \leq 2 \log \frac{|\mathcal{S}|T}{\delta}$ in the second line, which holds due to the choice of $T$. The following lemmas are required for deriving the upper bound, which hold for both cases of NTD and CTD.

**Lemma C.3.** *For any $t \in [T]$,*

$$\boldsymbol{\sigma}(\eta_t) - \boldsymbol{\sigma}(\eta) \leq 4 \|\Delta_t\|_{\bar{W}_1} \mathbf{1}.$$

**Lemma C.4.**

$$(\boldsymbol{I} - \sqrt{\gamma}\boldsymbol{P})^{-1} \boldsymbol{\sigma}(\eta) \leq \frac{4}{1-\gamma} \mathbf{1}.$$

Combining the upper bound with the two lemmas, we get the desired conclusion

$$\left( \left\| \sum_{k=1}^{t} \zeta_k(s) \right\| \right)_{s \in \mathcal{S}}$$

$$\leq 8 \sqrt{\frac{\left( \log^3 T \right) \left( \log \frac{|\mathcal{S}|T}{\delta} \right)}{(1-\gamma)T} \left[ 4 \max_{k: \, t/2 < k \leq t} \|\Delta_{k-1}\|_{\bar{W}_1} (\boldsymbol{I} - \sqrt{\gamma}\boldsymbol{P})^{-1} \mathbf{1} + \frac{8}{1-\gamma} \mathbf{1} \right]} + \frac{12 \left( \log^3 T \right) \left( \log \frac{|\mathcal{S}|T}{\delta} \right)}{(1-\gamma)^{3/2}T} \mathbf{1}$$

$$\leq 22 \sqrt{\frac{\left( \log^3 T \right) \left( \log \frac{|\mathcal{S}|T}{\delta} \right)}{(1-\gamma)^2 T} \left( 1 + \max_{k: \, t/2 < k \leq t} \|\Delta_{k-1}\|_{\bar{W}_1} \right)} \mathbf{1} + \frac{12 \left( \log^3 T \right) \left( \log \frac{|\mathcal{S}|T}{\delta} \right)}{(1-\gamma)^{3/2}T} \mathbf{1}$$

$$\leq 34 \sqrt{\frac{\left( \log^3 T \right) \left( \log \frac{|\mathcal{S}|T}{\delta} \right)}{(1-\gamma)^2 T} \left( 1 + \max_{k: \, t/2 < k \leq t} \|\Delta_{k-1}\|_{\bar{W}_1} \right)} \mathbf{1},$$

$$(44)$$

where in the last line, we used that, excluding the constant term, the first term is larger than the second term, given the choice of $T \geq \frac{C_4 \log^3 T}{\varepsilon^2 (1-\gamma)^3} \log \frac{|\mathcal{S}|T}{\delta}$.

$\square$

## C.4 Solve the Recurrence Relation

**Theorem C.1.** *Suppose for all $t \geq \frac{T}{c_6 \log T}$,*

$$\|\Delta_t\|_{\bar{W}_1} \leq 35 \sqrt{\frac{\left( \log^3 T \right) \left( \log \frac{|\mathcal{S}|T}{\delta} \right)}{(1-\gamma)^3 T} \left( 1 + \max_{k: \, t/2 < k \leq t} \|\Delta_{k-1}\|_{\bar{W}_1} \right)}.$$

*Then there exists some large universal constant $C_7 > 0$, such that*

$$\|\Delta_T\|_{\bar{W}_1} \leq C_7 \left( \sqrt{\frac{\left( \log^3 T \right) \left( \log \frac{|\mathcal{S}|T}{\delta} \right)}{(1-\gamma)^3 T}} + \frac{\left( \log^3 T \right) \left( \log \frac{|\mathcal{S}|T}{\delta} \right)}{(1-\gamma)^3 T} \right).$$

*Proof.* For any $k \geq 0$, we denote

$$u_k := \max \left\{ \|\Delta_t\|_{\bar{W}_1} \; \Big| \; 2^k \frac{T}{c_6 \log T} \leq t \leq T \right\}, \tag{45}$$

for $0 \leq k \leq \log_2 (c_6 \log T)$. We can see that $\|\Delta_T\|_{\bar{W}_1} \leq u_k$ for any valid $k$. Hence, it suffices to show the upper bound holds for $u_k$ for any valid $k$. It can be verified that $u_0 \leq \frac{1}{1-\gamma}$, and for $k \geq 0$

$$u_{k+1} \leq 35 \sqrt{\frac{\left(\log^3 T\right) \left(\log \frac{|\mathcal{S}|T}{\delta}\right)}{(1-\gamma)^3 T}} (1 + u_k). \tag{46}$$

We first show that once $u_k \leq 1$, the subsequent values of $u_{k+l}$ will also remain upper bounded by $1$. Namely, if $u_k \leq 1$ for some $k \geq 1$, then

$$u_{k+1} \leq 35 \sqrt{\frac{2 \left(\log^3 T\right) \left(\log \frac{|\mathcal{S}|T}{\delta}\right)}{(1-\gamma)^3 T}} \leq 1, \tag{47}$$

if $T \geq \frac{2450 \log^3 T \log \frac{|\mathcal{S}|T}{\delta}}{(1-\gamma)^3}$.

Let $\tau := \inf \{k : u_k \leq 1\}$, then for any $k > \tau$, we have

$$u_k \leq 35 \sqrt{\frac{2 \left(\log^3 T\right) \left(\log \frac{|\mathcal{S}|T}{\delta}\right)}{(1-\gamma)^3 T}} =: a. \tag{48}$$

For $k \leq \tau$, we have $u_k \geq 1$ and thereby

$$u_{k+1} \leq 35 \sqrt{\frac{2 \left(\log^3 T\right) \left(\log \frac{|\mathcal{S}|T}{\delta}\right)}{(1-\gamma)^3 T}} u_k = a\sqrt{u_k}, \tag{49}$$

*i.e.,*

$$\log u_{k+1} - 2 \log a \leq \frac{1}{2} \left(\log u_k - 2 \log a\right). \tag{50}$$

Apply it recursively, we have

$$\log u_{k+1} \leq 2 \log a + \left(\frac{1}{2}\right)^{k+1} \left(\log u_0 - 2 \log a\right), \tag{51}$$

*i.e.,*

$$u_{k+1} \leq a^2 \left(\frac{u_0}{a^2}\right)^{1/2^k} = a^{2\left(1-1/2^k\right)} u_0^{1/2^k} \leq a^{2\left(1-1/2^k\right)} \frac{1}{(1-\gamma)^{1/2^k}}. \tag{52}$$

To sum up, for any $k \geq 0$, $u_{k+1}$ is always less than the sum of the upper bounds in cases of $k > \tau$ and $k \leq \tau$,

$$u_{k+1} \leq a + a^{2\left(1-1/2^k\right)} \frac{1}{(1-\gamma)^{1/2^k}} \tag{53}$$

Note that, $a^{2\left(1-1/2^k\right)} \leq \max\{a, \sqrt{a}\}$, and if we take $k \geq c_8 \log\log \frac{1}{1-\gamma}$ for any constant $c_8$, we have $\frac{1}{(1-\gamma)^{1/2^k}} = O(1)$. We can take the constant $c_8$ small enough such that $c_8 \log\log \frac{1}{1-\gamma} < \log_2 (c_6 \log T)$ (this can be done and $c_8$ is universal since $\frac{1}{1-\gamma} = o(T)$), and thereby we can find a valid $k^\star \geq c_8 \log\log \frac{1}{1-\gamma} + 1$. Then

$$\|\Delta_T\|_{\bar{W}_1} \leq u_{k^\star} \leq C_7 \left( \sqrt{\frac{\left(\log^3 T\right) \left(\log \frac{|\mathcal{S}|T}{\delta}\right)}{(1-\gamma)^3 T}} + \frac{\left(\log^3 T\right) \left(\log \frac{|\mathcal{S}|T}{\delta}\right)}{(1-\gamma)^3 T} \right), \tag{54}$$

which is the desired conclusion, and $C_7$ is some large universal constant related to $c_8$. $\qquad \square$

## C.5 Analysis of Corollaries 4.1 and 4.2

The difference in the proof compared to Section 5.2 arises in Lemma 5.2 when we control term (II). Now we further bound the result in Lemma C.3 by the Cramér norm of the error term,

$$\boldsymbol{\sigma}(\eta_t) - \boldsymbol{\sigma}(\eta) \leq 4 \, \|\Delta_t\|_{\bar{W}_1} \, \mathbf{1} \leq \frac{1}{\sqrt{1-\gamma}} \, \|\Delta_t\| \, \mathbf{1}. \tag{55}$$

In the same way, we can derive the following recurrence relation: with probability at least $1 - \delta$, for all $t \geq \frac{T}{c_6 \log T}$

$$\|\Delta_t\| \leq 35 \sqrt{\frac{\left(\log^3 T\right) \left(\log \frac{|\mathcal{S}|T}{\delta}\right)}{(1-\gamma)^{5/2}T}} \left(1 + \max_{k:\, t/2 < k \leq t} \|\Delta_{k-1}\|\right). \tag{56}$$

By repeating the reasoning of Theorem C.1, we can obtain the desired conclusion,

$$\|\Delta_T\| \leq C_7 \left( \sqrt{\frac{\left(\log^3 T\right) \left(\log \frac{|\mathcal{S}|T}{\delta}\right)}{(1-\gamma)^{5/2}T}} + \frac{\left(\log^3 T\right) \left(\log \frac{|\mathcal{S}|T}{\delta}\right)}{(1-\gamma)^{5/2}T} \right), \tag{57}$$

which is less than $\varepsilon$ if we take $C_4 \geq 2C_7^2$ and $T \geq \frac{C_4 \log^3 T}{\varepsilon^2 (1-\gamma)^{5/2}} \log \frac{|\mathcal{S}|T}{\delta}$. Here, $C_7 > 1$ is a large universal constant depending on $c_6$.

## C.6 Proof of Lemma C.1

*Proof.* We first introduce some notations. For any matrix of operators $\mathcal{U} \in \mathcal{L}(\mathcal{M})^{\mathcal{S} \times \mathcal{S}}$, we denote $\mathcal{U}(s) = (\mathcal{U}(s, s'))_{s' \in \mathcal{S}} \in \mathcal{L}(\mathcal{M})^{\mathcal{S}}$ as the $s$-row of $\mathcal{U}$. And for any $\xi \in \mathcal{M}^{\mathcal{S}}$, we define the vector inner product operation $\mathcal{U}(s)\xi := \sum_{s' \in \mathcal{S}} \mathcal{U}(s, s')\xi(s') \in \mathcal{M}$.

We need the following lemma, which holds for both cases of NTD and CTD.

**Lemma C.5.** *For any $\nu \in \mathcal{M}$, $n \in \mathbb{N}$, $(\alpha_i)_{i \in [n]} \in [0,1]^n$, let $\mathcal{U}_n = \prod_{i=1}^n [(1-\alpha_i)\mathcal{I} + \alpha_i \mathcal{T}]$, $\boldsymbol{U}_n = \prod_{i=1}^n [(1-\alpha_i)\boldsymbol{I} + \alpha_i\sqrt{\gamma}\boldsymbol{P}]$, $u_n = \prod_{i=1}^n [(1-\alpha_i) + \alpha_i\sqrt{\gamma}]$ then for any $s, s' \in \mathcal{S}$, we have*

$$\|\mathcal{U}_n(s, s')\nu\|^2 \leq u_n \boldsymbol{U}_n(s, s') \|\nu\|^2.$$

Utilizing this lemma, we get the following result. Recall that $\widehat{\mathcal{T}}$ is a random operator and has the same distribution as $\mathcal{T}_1$. Then, for any non-random $\xi \in \mathcal{M}^{\mathcal{S}}$,

$$\mathbb{E}\left[\left\|\mathcal{U}_n(s)(\widehat{\mathcal{T}} - \mathcal{T})\xi\right\|^2\right]$$

$$= \mathbb{E}\left[\left\|\sum_{s' \in \mathcal{S}} \mathcal{U}_n(s, s') \left[(\widehat{\mathcal{T}} - \mathcal{T})\xi\right](s')\right\|^2\right]$$

$$= \mathbb{E}\left[\left\|\sum_{s' \in \mathcal{S}} \mathcal{U}_n(s, s') \left[\widehat{\mathcal{T}}(s')\xi - \mathcal{T}(s')\xi\right]\right\|^2\right]$$

$$= \sum_{s' \in \mathcal{S}} \mathbb{E}\left[\left\|\mathcal{U}_n(s, s') \left[\widehat{\mathcal{T}}(s')\xi - \mathcal{T}(s')\xi\right]\right\|^2\right] \tag{58}$$

$$\leq u_n \sum_{s' \in \mathcal{S}} \boldsymbol{U}_n(s, s')\mathbb{E}\left[\left\|\widehat{\mathcal{T}}(s')\xi - \mathcal{T}(s')\xi\right\|^2\right]$$

$$= u_n \sum_{s' \in \mathcal{S}} \boldsymbol{U}_n(s, s')\boldsymbol{\sigma}(\xi)(s')$$

$$= u_n \boldsymbol{U}_n(s)\boldsymbol{\sigma}(\xi),$$

where we used different rows of $\widehat{\mathcal{T}}$ are independent, and $\widehat{\mathcal{T}}(s')\xi$ is an unbiased estimator of $\mathcal{T}(s')\xi \in \mathcal{M}$. Hence, $\text{Var}\left(\mathcal{U}_n(\widehat{\mathcal{T}} - \mathcal{T})\xi\right) \leq u_n \boldsymbol{U}_n \boldsymbol{\sigma}(\xi)$.

Now, we are ready to bound $\text{Var}_{k-1}\left(\zeta_k\right)$

$$\text{Var}_{k-1}\left(\zeta_k\right) = \alpha_k^2 \text{Var}_{k-1}\left(\prod_{i=k+1}^{t}\left[(1-\alpha_i)\mathcal{I} + \alpha_i\mathcal{T}\right]\left(\mathcal{T}_k - \mathcal{T}\right)\eta_{k-1}\right)$$

$$\leq \alpha_k^2 \prod_{i=k+1}^{t}\left[(1-\alpha_i) + \alpha_i\sqrt{\gamma}\right]\prod_{i=k+1}^{t}\left[(1-\alpha_i)\boldsymbol{I} + \alpha_i\sqrt{\gamma}\boldsymbol{P}\right]\boldsymbol{\sigma}(\eta_{k-1}) \quad (59)$$

$$= \alpha_k\beta_k^{(t)}\prod_{i=k+1}^{t}\left[(1-\alpha_i)\boldsymbol{I} + \alpha_i\sqrt{\gamma}\boldsymbol{P}\right]\boldsymbol{\sigma}(\eta_{k-1}).$$

$\square$

## C.7 Proof of Lemma C.2

*Proof.*

$$\sum_{k=t/2+1}^{t}\alpha_k\prod_{i=k+1}^{t}\left[(1-\alpha_i)\boldsymbol{I} + \alpha_i\sqrt{\gamma}\boldsymbol{P}\right]$$

$$= \sum_{k=t/2+1}^{t}\prod_{i=k+1}^{t}\left[(1-\alpha_i)\boldsymbol{I} + \alpha_i\sqrt{\gamma}\boldsymbol{P}\right]\alpha_k(\boldsymbol{I} - \sqrt{\gamma}\boldsymbol{P})(\boldsymbol{I} - \sqrt{\gamma}\boldsymbol{P})^{-1}$$

$$= \sum_{k=t/2+1}^{t}\left\{\prod_{i=k+1}^{t}\left[(1-\alpha_i)\boldsymbol{I} + \alpha_i\sqrt{\gamma}\boldsymbol{P}\right] - \prod_{i=k}^{t}\left[(1-\alpha_i)\boldsymbol{I} + \alpha_i\sqrt{\gamma}\boldsymbol{P}\right]\right\}(\boldsymbol{I} - \sqrt{\gamma}\boldsymbol{P})^{-1} \quad (60)$$

$$= \left\{\boldsymbol{I} - \prod_{i=t/2+1}^{t}\left[(1-\alpha_i)\boldsymbol{I} + \alpha_i\sqrt{\gamma}\boldsymbol{P}\right]\right\}(\boldsymbol{I} - \sqrt{\gamma}\boldsymbol{P})^{-1}$$

$$\leq (\boldsymbol{I} - \sqrt{\gamma}\boldsymbol{P})^{-1},$$

where the inequality holds entry-wise since we can verify that all entries of $(\boldsymbol{I} - \sqrt{\gamma}\boldsymbol{P})^{-1} = \sum_{k=0}^{\infty}\left(\sqrt{\gamma}\boldsymbol{P}\right)^k$ and $(1-\alpha_i)\boldsymbol{I} + \alpha_i\sqrt{\gamma}\boldsymbol{P}$ are non-negative. $\square$

## C.8 Proof of Lemma C.3

*Proof.* For any $s \in \mathcal{S}$,

$$\boldsymbol{\sigma}(\eta_t)(s) - \boldsymbol{\sigma}(\eta)(s)$$

$$= \int_0^{\frac{1}{1-\gamma}}\left\{\mathbb{E}\left[F_{(\widehat{\mathcal{T}}\eta_t)(s)}^2(x)\right] - F_{(\mathcal{T}\eta_t)(s)}^2(x) - \mathbb{E}\left[F_{(\widehat{\mathcal{T}}\eta)(s)}^2(x)\right] + F_{(\mathcal{T}\eta)(s)}^2(x)\right\}dx$$

$$= \int_0^{\frac{1}{1-\gamma}}\left\{\mathbb{E}\left[F_{(\widehat{\mathcal{T}}\eta_t)(s)}^2(x) - F_{(\widehat{\mathcal{T}}\eta)(s)}^2(x)\right] + F_{(\mathcal{T}\eta)(s)}^2(x) - F_{(\mathcal{T}\eta_t)(s)}^2(x)\right\}dx$$

$$= \int_0^{\frac{1}{1-\gamma}}\left\{\mathbb{E}\left[\left(F_{(\widehat{\mathcal{T}}\eta_t)(s)}(x) - F_{(\widehat{\mathcal{T}}\eta)(s)}(x)\right)\left(F_{(\widehat{\mathcal{T}}\eta_t)(s)}(x) + F_{(\widehat{\mathcal{T}}\eta)(s)}(x)\right)\right]\right. \quad (61)$$

$$\left. + \left(F_{(\mathcal{T}\eta)(s)}(x) - F_{(\mathcal{T}\eta_t)(s)}(x)\right)\left(F_{(\mathcal{T}\eta)(s)}(x) + F_{(\mathcal{T}\eta_t)(s)}(x)\right)\right\}dx$$

$$\leq 2\int_0^{\frac{1}{1-\gamma}}\left\{\mathbb{E}\left[\left|F_{(\widehat{\mathcal{T}}\eta_t)(s)}(x) - F_{(\widehat{\mathcal{T}}\eta)(s)}(x)\right|\right] + \left|F_{(\mathcal{T}\eta)(s)}(x) - F_{(\mathcal{T}\eta_t)(s)}(x)\right|\right\}dx$$

$$= 2\left(\mathbb{E}\left[\left\|\widehat{\mathcal{T}}\left(\eta_t - \eta\right)(s)\right\|_{W_1}\right] + \left\|\mathcal{T}\left(\eta_t - \eta\right)(s)\right\|_{W_1}\right).$$

In the case of NTD, $\mathcal{T}$ and $\widehat{\mathcal{T}}$ are $\gamma$-contraction w.r.t. the supreme 1-Wasserstein metric, hence

$$
\begin{aligned}
\boldsymbol{\sigma}(\eta_t)(s) - \boldsymbol{\sigma}(\eta)(s) &\leq 2 \left( \mathbb{E}\left[ \left\| \widehat{\mathcal{T}}(\eta_t - \eta)(s) \right\|_{W_1} \right] + \left\| \mathcal{T}(\eta_t - \eta)(s) \right\|_{W_1} \right) \\
&\leq 4\gamma \left\| \eta_t - \eta \right\|_{\bar{W}_1} \\
&\leq 4 \left\| \Delta_t \right\|_{\bar{W}_1}.
\end{aligned}
\tag{62}
$$

In the case of CTD, if we can show $\Pi_K$ is non-expansive w.r.t. 1-Wasserstein metric, the conclusion still holds. For any $x, y \in \left[ 0, \frac{1}{1-\gamma} \right]$ such that $x < y$, we denote $x \in [x_k, x_{k+1})$ and $y \in [x_l, x_{l+1})$, then $k \leq l$, by the definition of $\Pi_K$, we have

$$
\Pi_K(\delta_x) = \frac{x_{k+1} - y}{\iota_K} \delta_{x_k} + \frac{y - x_k}{\iota_K} \delta_{x_{k+1}},
\tag{63}
$$

$$
\Pi_K(\delta_y) = \frac{x_{l+1} - y}{\iota_K} \delta_{x_l} + \frac{y - x_l}{\iota_K} \delta_{x_{l+1}}.
\tag{64}
$$

If $k = l$, we can check that $W_1(\Pi_K \delta_x, \Pi_K \delta_y) = \iota_K \frac{y-x}{\iota_K} = y - x$. If $k < l$, we have $W_1(\Pi_K \delta_x, \Pi_K \delta_y) \leq W_1(\Pi_K \delta_x, x_{k+1}) + W_1(x_{k+1}, x_l) + W_1(x_l, \Pi_K \delta_y) = (x_{k+1} - x) + (x_l - x_{k+1}) + (y - x_{x_l}) = y - x$. Hence, for any $\nu_1, \nu_2 \in \mathscr{P}$ and for any transport plan $\kappa \in \Gamma(\nu_1, \nu_2)$, the previous results tell us the cost of the transport plan $\Pi_K \kappa \in \Gamma(\Pi_K \nu_1, \Pi_K \nu_2)$ induced by $\Pi_K$ is no greater than the cost of $\kappa$. Consequently, $W_1(\Pi_K \nu_1, \Pi_K \nu_2) \leq W_1(\nu_1, \nu_2)$, *i.e.*, $\Pi_K$ is non-expansive w.r.t. 1-Wasserstein metric, which is desired. $\square$

### C.9 Proof of Lemma C.4

*Proof.* Firstly, we show that for any $\boldsymbol{v} \geq \boldsymbol{0}$, we have $\left\| (\boldsymbol{I} - \sqrt{\gamma}\boldsymbol{P})^{-1} \boldsymbol{v} \right\| \leq 2 \left\| (\boldsymbol{I} - \gamma\boldsymbol{P})^{-1} \boldsymbol{v} \right\|$

$$
\begin{aligned}
\left\| (\boldsymbol{I} - \sqrt{\gamma}\boldsymbol{P})^{-1} \boldsymbol{v} \right\| &= \left\| (\boldsymbol{I} - \sqrt{\gamma}\boldsymbol{P})^{-1}(\boldsymbol{I} - \gamma\boldsymbol{P})(\boldsymbol{I} - \gamma\boldsymbol{P})^{-1} \boldsymbol{v} \right\| \\
&= \left\| (\boldsymbol{I} - \sqrt{\gamma}\boldsymbol{P})^{-1} \left[ (1 - \sqrt{\gamma})\boldsymbol{I} + \sqrt{\gamma}(\boldsymbol{I} - \sqrt{\gamma}\boldsymbol{P}) \right] (\boldsymbol{I} - \gamma\boldsymbol{P})^{-1} \boldsymbol{v} \right\| \\
&= \left\| \left[ (1 - \sqrt{\gamma})(\boldsymbol{I} - \sqrt{\gamma}\boldsymbol{P})^{-1} + \sqrt{\gamma}\boldsymbol{I} \right] (\boldsymbol{I} - \gamma\boldsymbol{P})^{-1} \boldsymbol{v} \right\| \\
&\leq (1 - \sqrt{\gamma}) \left\| (\boldsymbol{I} - \sqrt{\gamma}\boldsymbol{P})^{-1}(\boldsymbol{I} - \gamma\boldsymbol{P})^{-1} \boldsymbol{v} \right\| + \sqrt{\gamma} \left\| (\boldsymbol{I} - \gamma\boldsymbol{P})^{-1} \boldsymbol{v} \right\| \\
&\leq \left( \frac{1 - \sqrt{\gamma}}{1 - \sqrt{\gamma}} + \sqrt{\gamma} \right) \left\| (\boldsymbol{I} - \gamma\boldsymbol{P})^{-1} \boldsymbol{v} \right\| \\
&\leq 2 \left\| (\boldsymbol{I} - \gamma\boldsymbol{P})^{-1} \boldsymbol{v} \right\|.
\end{aligned}
\tag{65}
$$

In the case of NTD, by Corollary D.1, we have

$$
\left\| (\boldsymbol{I} - \gamma\boldsymbol{P})^{-1} \boldsymbol{\sigma}(\eta) \right\| \leq \frac{1}{1 - \gamma},
\tag{66}
$$

In the case of CTD, by Corollary 5.12 in [Rowland et al., 2024b], we have

$$
\left\| (\boldsymbol{I} - \gamma\boldsymbol{P})^{-1} \boldsymbol{\sigma}(\eta) \right\| \leq \frac{2}{1 - \gamma},
\tag{67}
$$

given $K > \frac{4}{1-\gamma}$. $\square$

### C.10 Proof of Lemma C.5

*Proof.* We proof this result by induction. For $n = 0$, we have $\mathcal{U}_0 = \mathcal{I}$, $\boldsymbol{U}_0 = \boldsymbol{I}$, $u_0 = 1$, thereby the inequality holds trivially. Suppose the inequality holds true for $n - 1$. To prove that the inequality holds for $n$, it is sufficient to show that, for any $\mu \in \mathcal{M}$,

$$
\left\| \left[ (1 - \alpha_n)\delta_{s,s'} + \alpha_n \mathcal{T}(s, s') \right] \mu \right\|^2 \leq \left[ (1 - \alpha_n) + \alpha_n \sqrt{\gamma} \right] \left[ (1 - \alpha_n)\delta_{s,s'} + \alpha_n \sqrt{\gamma}\boldsymbol{P}(s, s') \right] \left\| \mu \right\|^2,
$$

where $\delta_{s,s'} = 1$ if $s = s'$, and $0$ otherwise.

LHS can be bounded as follow

$$\left\| \left[ (1 - \alpha_n)\delta_{s,s'} + \alpha_n \mathcal{T}(s,s') \right] \mu \right\|^2$$

$$= (1 - \alpha_n)^2 \delta_{s,s'} \|\mu\|^2 + 2(1 - \alpha_n)\alpha_n \delta_{s,s'} \langle \mu, \mathcal{T}(s,s')\mu \rangle + \alpha_n^2 \|\mathcal{T}(s,s')\mu\|^2 \tag{68}$$

$$\leq (1 - \alpha_n)^2 \delta_{s,s'} \|\mu\|^2 + 2(1 - \alpha_n)\alpha_n \delta_{s,s'} \|\mu\| \|\mathcal{T}(s,s')\mu\| + \alpha_n^2 \|\mathcal{T}(s,s')\mu\|^2,$$

where we used Cauchy-Schwarz inequality. We need to give an upper bound for $\|\mathcal{T}(s,s')\mu\|^2$.

Note that $(\Pi_K \mathcal{T}^\pi)(s,s') = \Pi_K(\mathcal{T}^\pi(s,s'))$ and $\|\Pi_K\| = 1$, we only need to consider the case of NTD, by the definition of $\mathcal{T}(s,s')$, we have

$$\|\mathcal{T}(s,s')\mu\|^2 = \int_0^{\frac{1}{1-\gamma}} \left[ \sum_{a \in \mathcal{A}} \pi(a|s)P(s'|s,a) \int_0^1 F_\mu \left( \frac{x-r}{\gamma} \right) \mathcal{P}_R(dr|s,a) \right]^2 dx$$

$$= \boldsymbol{P}(s,s')^2 \int_0^{\frac{1}{1-\gamma}} \left[ \sum_{a \in \mathcal{A}} \frac{\pi(a|s)P(s'|s,a)}{\boldsymbol{P}(s,s')} \int_0^1 F_\mu \left( \frac{x-r}{\gamma} \right) \mathcal{P}_R(dr|s,a) \right]^2 dx$$

$$= \boldsymbol{P}(s,s')^2 \int_0^{\frac{1}{1-\gamma}} \left\{ \mathbb{E}_{a \sim \pi(\cdot|s), r \sim \mathcal{P}_R(\cdot|s,a)} \left[ F_\mu \left( \frac{x-r}{\gamma} \right) \Big| s' \right] \right\}^2 dx \tag{69}$$

$$\leq \boldsymbol{P}(s,s')^2 \mathbb{E}_{a \sim \pi(\cdot|s), r \sim \mathcal{P}_R(\cdot|s,a)} \left\{ \int_0^{\frac{1}{1-\gamma}} \left[ F_\mu \left( \frac{x-r}{\gamma} \right) \right]^2 dx \Big| s' \right\}$$

$$= \gamma \boldsymbol{P}(s,s')^2 \|\mu\|^2,$$

where we used Jensen's inequality and Fubini's theorem. Substitute it back to the upper bound,

$$\left\| \left[ (1 - \alpha_n)\delta_{s,s'} + \alpha_n \mathcal{T}(s,s') \right] \mu \right\|^2$$

$$\leq (1 - \alpha_n)^2 \delta_{s,s'} \|\mu\|^2 + 2(1 - \alpha_n)\alpha_n \delta_{s,s'} \|\mu\| \|\mathcal{T}(s,s')\mu\| + \alpha_n^2 \|\mathcal{T}(s,s')\mu\|^2$$

$$\leq \left[ (1 - \alpha_n)^2 \delta_{s,s'} + 2(1 - \alpha_n)\alpha_n \delta_{s,s'} \sqrt{\gamma} \boldsymbol{P}(s,s') + \alpha_n^2 \gamma \boldsymbol{P}(s,s')^2 \right] \|\mu\|^2 \tag{70}$$

$$= \left[ (1 - \alpha_n)^2 \delta_{s,s'} + \alpha_n \sqrt{\gamma} \boldsymbol{P}(s,s') \right]^2 \|\mu\|^2$$

$$\leq \left[ (1 - \alpha_n) + \alpha_n \sqrt{\gamma} \right] \left[ (1 - \alpha_n)\delta_{s,s'} + \alpha_n \sqrt{\gamma} \boldsymbol{P}(s,s') \right] \|\mu\|^2,$$

which is desired. $\qquad\square$

## D Stochastic Distributional Bellman Equation and Operator

In this section, we use the same notations as in Appendix C and only consider the NTD setting. Inspired by stochastic categorical CDF Bellman operator introduced in [Rowland et al., 2024b], we introduce stochastic distributional Bellman operator $\mathscr{T}: \Delta\left(\mathscr{P}^{\mathcal{S}}\right) \to \Delta\left(\mathscr{P}^{\mathcal{S}}\right)$ to derive an upper bound for $\left\|(\boldsymbol{I} - \gamma \boldsymbol{P})^{-1} \sigma(\eta)\right\|$ in the case of NTD. For any $\phi \in \Delta\left(\mathscr{P}^{\mathcal{S}}\right)$, we denote $\boldsymbol{\eta}_\phi$ be the random element in $\mathscr{P}^{\mathcal{S}}$ with law $\phi$.

$$\mathscr{T}\phi := \text{Law}\left(\widehat{\mathcal{T}}\boldsymbol{\eta}_\phi\right), \tag{71}$$

where $(\widehat{\mathcal{T}}\boldsymbol{\eta}_\phi)(\omega) := (\widehat{\mathcal{T}})(\omega)(\boldsymbol{\eta}_\phi)(\omega) \in \mathscr{P}^{\mathcal{S}}$ for any $\omega \in \Omega$, $\Omega$ is the corresponding probability space, and $\widehat{\mathcal{T}}$ is independent of $\boldsymbol{\eta}_\phi$. In this part, $\widehat{\mathcal{T}}$ does not consist of $\Pi_K$ since we only consider the NTD setting.

We consider the 1-Wasserstein metric $W_1$ on $\Delta\left(\mathscr{P}^{\mathcal{S}}\right)$, the space of all probability measures on the space $\left(\mathscr{P}^{\mathcal{S}}, \bar{\ell}_2\right)$. Since $\left(\mathscr{P}^{\mathcal{S}}, \bar{\ell}_2\right)$ is Polish (complete and separable), the space $\left(\Delta\left(\mathscr{P}^{\mathcal{S}}\right), W_1\right)$ is also Polish (Theorem 6.18 in [Villani et al., 2009]).

**Proposition D.1.** *The stochastic distributional Bellman operator $\mathscr{T}$ is a $\sqrt{\gamma}$-contraction on $\Delta\left(\mathscr{P}^{\mathcal{S}}\right)$, i.e., for any $\phi, \phi' \in \Delta\left(\mathscr{P}^{\mathcal{S}}\right)$, we have*

$$W_1\left(\mathscr{T}\phi, \mathscr{T}\phi'\right) \leq \sqrt{\gamma} W_1\left(\phi, \phi'\right).$$

*Proof.* Let $\boldsymbol{\kappa}^\star \in \Gamma(\boldsymbol{\phi}, \boldsymbol{\phi}')$ be the optimal coupling between $\boldsymbol{\phi}$ and $\boldsymbol{\phi}'$. The existence of $\boldsymbol{\kappa}^\star$ is guaranteed by Theorem 4.1 in [Villani et al., 2009]. And let the random element $\boldsymbol{\xi} = (\boldsymbol{\xi}_1, \boldsymbol{\xi}_2)$ in $\left(\mathscr{P}^{\mathcal{S}}\right)^2$ has the law $\boldsymbol{\kappa}^\star$, where $\boldsymbol{\xi}_1$ and $\boldsymbol{\xi}_2$ are both random elements in $\mathscr{P}^{\mathcal{S}}$. We denote $\mathscr{T}\boldsymbol{\kappa}^\star :=$ $\text{Law}\left[\left(\widehat{\mathcal{T}}\boldsymbol{\xi}_1, \widehat{\mathcal{T}}\boldsymbol{\xi}_2\right)\right] \in \Gamma(\mathscr{T}\boldsymbol{\phi}, \mathscr{T}\boldsymbol{\phi}')$.

$$
\begin{aligned}
W_1\left(\mathscr{T}\boldsymbol{\phi}, \mathscr{T}\boldsymbol{\phi}'\right) &= \inf_{\boldsymbol{\kappa} \in \Gamma(\mathscr{T}\boldsymbol{\phi}, \mathscr{T}\boldsymbol{\phi}')} \int_{(\mathscr{P}^{\mathcal{S}})^2} \bar{\ell}_2\left(\xi, \xi'\right) \boldsymbol{\kappa}\left(d\xi, d\xi'\right) \\
&\leq \int_{(\mathscr{P}^{\mathcal{S}})^2} \bar{\ell}_2\left(\xi, \xi'\right) \mathscr{T}\boldsymbol{\kappa}^\star\left(d\xi, d\xi'\right) \\
&= \mathbb{E}\left[\bar{\ell}_2\left(\widehat{\mathcal{T}}\boldsymbol{\xi}_1, \widehat{\mathcal{T}}\boldsymbol{\xi}_2\right)\right] \\
&\leq \sqrt{\gamma}\mathbb{E}\left[\bar{\ell}_2\left(\boldsymbol{\xi}_1, \boldsymbol{\xi}_2\right)\right] \\
&= \sqrt{\gamma}\int_{(\mathscr{P}^{\mathcal{S}})^2} \bar{\ell}_2\left(\xi, \xi'\right) \boldsymbol{\kappa}^\star\left(d\xi, d\xi'\right) \\
&= \sqrt{\gamma}\inf_{\boldsymbol{\kappa} \in \Gamma(\boldsymbol{\phi}, \boldsymbol{\phi}')} \int_{(\mathscr{P}^{\mathcal{S}})^2} \bar{\ell}_2\left(\xi, \xi'\right) \boldsymbol{\kappa}\left(d\xi, d\xi'\right) \\
&= \sqrt{\gamma}W_1\left(\boldsymbol{\phi}, \boldsymbol{\phi}'\right).
\end{aligned}
\tag{72}
$$

$\square$

By the proposition and contraction mapping theorem, there exists a unique fixed point of $\mathscr{T}$, we denote $\boldsymbol{\psi} \in \Delta\left(\mathscr{P}^{\mathcal{S}}\right)$ as the fixed point. Hence, the stochastic distributional Bellman equation reads

$$
\boldsymbol{\psi} = \mathscr{T}\boldsymbol{\psi}.
\tag{73}
$$

We denote $\boldsymbol{\eta}_{\boldsymbol{\psi}}$ as the random element in $\mathscr{P}$ with law $\boldsymbol{\psi}$, then $\widehat{\mathcal{T}}\boldsymbol{\eta}_{\boldsymbol{\psi}}$ and $\boldsymbol{\eta}_{\boldsymbol{\psi}}$ have the same law. As shown in the following proposition, $\boldsymbol{\eta}_{\boldsymbol{\psi}}$ can be regarded as a noisy version of $\eta$.

**Proposition D.2.**

$$
\mathbb{E}\left[\boldsymbol{\eta}_{\boldsymbol{\psi}}\right] = \eta,
$$

*where the expectation is regarded as the Bochner integral in the space of all finite measures on $\mathscr{P}^{\mathcal{S}}$, which is a normed linear space equipped with Cramér metric as its norm.*

*Proof.*

$$
\begin{aligned}
\mathbb{E}\left[\boldsymbol{\eta}_{\boldsymbol{\psi}}\right] &= \mathbb{E}\left[\widehat{\mathcal{T}}\boldsymbol{\eta}_{\boldsymbol{\psi}}\right] \\
&= \mathbb{E}\left\{\mathbb{E}\left[\widehat{\mathcal{T}}\boldsymbol{\eta}_{\boldsymbol{\psi}}\Big|\boldsymbol{\eta}_{\boldsymbol{\psi}}\right]\right\} \\
&= \mathbb{E}\left[\mathcal{T}\boldsymbol{\eta}_{\boldsymbol{\psi}}\right] \\
&= \mathcal{T}\mathbb{E}\left[\boldsymbol{\eta}_{\boldsymbol{\psi}}\right],
\end{aligned}
\tag{74}
$$

where we used $\widehat{\mathcal{T}}$ is independent of $\boldsymbol{\eta}_{\boldsymbol{\psi}}$. Since $\mathbb{E}\left[\boldsymbol{\eta}_{\boldsymbol{\psi}}\right]$ is the fixed point of $\mathcal{T}$, we have $\mathbb{E}\left[\boldsymbol{\eta}_{\boldsymbol{\psi}}\right] = \eta$. $\square$

Based on the concepts of $\mathscr{T}$ and $\boldsymbol{\psi}$, we can obtain the following second order distributional Bellman equation, which is similar to the classic second-order Bellman equation (Lemma 7 in [Gheshlaghi Azar et al., 2013]).

Recall the one-step Cramér variation $\boldsymbol{\sigma}(\eta) = \left(\mathbb{E}\left[\left\|\left(\widehat{\mathcal{T}}\eta\right)(s) - \eta(s)\right\|^2\right]\right)_{s \in \mathcal{S}} \in \mathbb{R}^{\mathcal{S}}$ used in the NTD setting. We denote $\boldsymbol{\sigma} := \boldsymbol{\sigma}(\eta)$ for simplicity, and $\boldsymbol{\Sigma} := \left(\mathbb{E}\left[\|\boldsymbol{\eta}_{\boldsymbol{\psi}}(s) - \eta(s)\|^2\right]\right)_{s \in \mathcal{S}} \in \mathbb{R}^{\mathcal{S}}$.

**Proposition D.3** (Second order distributional Bellman equation)**.**

$$
\boldsymbol{\Sigma} = \boldsymbol{\sigma} + \gamma \boldsymbol{P}\boldsymbol{\Sigma}.
$$

*Proof.* For any $s \in \mathcal{S}$,

$$
\begin{aligned}
\Sigma(s) &= \mathbb{E}\left[\|\boldsymbol{\eta}_{\boldsymbol{\psi}}(s) - \eta(s)\|^2\right] \\
&= \mathbb{E}\left[\left\|\left(\widehat{\mathcal{T}}\boldsymbol{\eta}_{\boldsymbol{\psi}}\right)(s) - \eta(s)\right\|^2\right] \\
&= \mathbb{E}\left[\left\|\left(\widehat{\mathcal{T}}\boldsymbol{\eta}_{\boldsymbol{\psi}}\right)(s) - \left(\widehat{\mathcal{T}}\eta\right)(s) + \left(\widehat{\mathcal{T}}\eta\right)(s) - \eta(s)\right\|^2\right] \\
&= \mathbb{E}\left[\left\|\left(\widehat{\mathcal{T}}\eta\right)(s) - \eta(s)\right\|^2\right] + \mathbb{E}\left[\left\|\left(\widehat{\mathcal{T}}\boldsymbol{\eta}_{\boldsymbol{\psi}}\right)(s) - \left(\widehat{\mathcal{T}}\eta\right)(s)\right\|^2\right],
\end{aligned}
\tag{75}
$$

where the last equality holds since the cross term is zero as below

$$
\begin{aligned}
&\mathbb{E}\left[\left\langle\left(\widehat{\mathcal{T}}\eta\right)(s) - \eta(s), \left(\widehat{\mathcal{T}}\boldsymbol{\eta}_{\boldsymbol{\psi}}\right)(s) - \left(\widehat{\mathcal{T}}\eta\right)(s)\right\rangle\right] \\
=&\mathbb{E}\left\{\mathbb{E}\left[\left\langle\left(\widehat{\mathcal{T}}\eta\right)(s) - \eta(s), \left(\widehat{\mathcal{T}}\boldsymbol{\eta}_{\boldsymbol{\psi}}\right)(s) - \left(\widehat{\mathcal{T}}\eta\right)(s)\right\rangle\Big|\widehat{\mathcal{T}}\right]\right\} \\
=&\mathbb{E}\left\{\left\langle\left(\widehat{\mathcal{T}}\eta\right)(s) - \eta(s), \mathbb{E}\left[\left(\widehat{\mathcal{T}}\boldsymbol{\eta}_{\boldsymbol{\psi}}\right)(s)\Big|\widehat{\mathcal{T}}\right] - \left(\widehat{\mathcal{T}}\eta\right)(s)\right\rangle\right\} \\
=&\mathbb{E}\left[\left\langle\left(\widehat{\mathcal{T}}\eta\right)(s) - \eta(s), \mathbf{0}\right\rangle\right] \\
=&0.
\end{aligned}
\tag{76}
$$

The first term in (75) is $\boldsymbol{\sigma}(s)$, we need to deal with the second term.

$$
\begin{aligned}
&\mathbb{E}\left[\left\|\left(\widehat{\mathcal{T}}\boldsymbol{\eta}_{\boldsymbol{\psi}}\right)(s) - \left(\widehat{\mathcal{T}}\eta\right)(s)\right\|^2\right] \\
=&\mathbb{E}\left\{\mathbb{E}\left[\left\|\left(\widehat{\mathcal{T}}\boldsymbol{\eta}_{\boldsymbol{\psi}}\right)(s) - \left(\widehat{\mathcal{T}}\eta\right)(s)\right\|^2\Big|\boldsymbol{\eta}_{\boldsymbol{\psi}}\right]\right\} \\
=&\mathbb{E}\left\{\mathbb{E}_{a(s)\sim\pi(\cdot|s),s'(s)\sim P(\cdot|s,a(s)),r(s)\sim\mathcal{P}_R(\cdot|s,a(s))}\left[\int_0^{\frac{1}{1-\gamma}}\left(F_{(\boldsymbol{\eta}_{\boldsymbol{\psi}})(s'(s))}\left(\frac{x-r}{\gamma}\right) - F_{\eta(s'(s))}\left(\frac{x-r}{\gamma}\right)\right)^2 dx\Big|\boldsymbol{\eta}_{\boldsymbol{\psi}}\right]\right\} \\
=&\gamma\sum_{s'\in\mathcal{S}}\mathbb{E}\left[\|\boldsymbol{\eta}_{\boldsymbol{\psi}}(s') - \eta(s')\|^2\right]\sum_{a\in\mathcal{A}}\pi(a|s)P(s'|s,a) \\
=&\gamma\sum_{s'\in\mathcal{S}}\boldsymbol{P}(s,s')\Sigma(s').
\end{aligned}
$$

$$\tag{77}$$

Put these together, and we can arrive at the conclusion. $\qquad\square$

Now, we can derive a tighter upper bound for $\left\|(\boldsymbol{I} - \gamma\boldsymbol{P})^{-1}\boldsymbol{\sigma}(\eta)\right\|$.

**Corollary D.1.**

$$
\left\|(\boldsymbol{I} - \gamma\boldsymbol{P})^{-1}\boldsymbol{\sigma}(\eta)\right\| \leq \left\|(\boldsymbol{I} - \gamma\boldsymbol{P})^{-1}\boldsymbol{\sigma}\right\| = \|\boldsymbol{\Sigma}\| \leq \frac{1}{1-\gamma}.
$$

*Proof.* Note that all entries of $(\boldsymbol{I}-\gamma\boldsymbol{P})^{-1} = \sum_{k=0}^{\infty}(\gamma\boldsymbol{P})^k$ are positive, thereby $(\boldsymbol{I}-\gamma\boldsymbol{P})^{-1}\boldsymbol{\sigma}(\eta) \leq (\boldsymbol{I}-\gamma\boldsymbol{P})^{-1}\boldsymbol{\sigma} = \boldsymbol{\Sigma}$, and $\boldsymbol{\Sigma}(s) = \mathbb{E}\left[\|\boldsymbol{\eta}_{\boldsymbol{\psi}}(s) - \eta(s)\|^2\right] \leq \int_0^{\frac{1}{1-\gamma}} dx = \frac{1}{1-\gamma}$ for any $s \in \mathcal{S}$. $\qquad\square$

# E   Other Technical Lemmas

**Lemma E.1** (Basic Inequalities for Metrics on the Space of Probability Measures). *For any* $\nu_1,\nu_2 \in \mathscr{P}$, *we have* $(\ell_2(\nu_1,\nu_2))^2 \leq W_1(\nu_1,\nu_2) \leq \frac{1}{\sqrt{1-\gamma}}\ell_2(\nu_1,\nu_2)$ *and* $W_p(\nu_1,\nu_2) \leq \frac{1}{(1-\gamma)^{1-\frac{1}{p}}}W_1^{\frac{1}{p}}(\nu_1,\nu_2)$.

*Proof.* By Cauchy-Schwarz inequality,

$$W_1(\nu_1, \nu_2)$$

$$= \int_0^{\frac{1}{1-\gamma}} |F_{\nu_1}(x) - F_{\nu_2}(x)| dx$$

$$\leq \sqrt{\int_0^{\frac{1}{1-\gamma}} 1^2 dx} \sqrt{\int_0^{\frac{1}{1-\gamma}} |F_{\nu_1}(x) - F_{\nu_2}(x)|^2 dx}$$

$$= \frac{1}{\sqrt{1-\gamma}} \ell_2(\nu_1, \nu_2).$$

And

$$\ell_2(\nu_1, \nu_2)$$

$$= \sqrt{\int_0^{\frac{1}{1-\gamma}} |F_{\nu_1}(x) - F_{\nu_2}(x)|^2 dx}$$

$$\leq \sqrt{\int_0^{\frac{1}{1-\gamma}} |F_{\nu_1}(x) - F_{\nu_2}(x)| dx}$$

$$= \sqrt{W_1(\nu_1, \nu_2)}.$$

And

$$W_p(\nu_1, \nu_2)$$

$$= \left( \inf_{\kappa \in \Gamma(\nu_1, \nu_2)} \int_{\left[0, \frac{1}{1-\gamma}\right]^2} |x - y|^p \, \kappa(dx, dy) \right)^{1/p}$$

$$\leq \frac{1}{(1-\gamma)^{1-\frac{1}{p}}} \left( \inf_{\kappa \in \Gamma(\nu_1, \nu_2)} \int_{\left[0, \frac{1}{1-\gamma}\right]^2} |x - y| \, \kappa(dx, dy) \right)^{1/p}$$

$$= \frac{1}{(1-\gamma)^{1-\frac{1}{p}}} W_1^{\frac{1}{p}}(\nu_1, \nu_2).$$

$\square$

**Lemma E.2.** $\left( \mathcal{M}, \|\cdot\|_{\ell_2} \right)$ and $\left( \mathcal{M}, \|\cdot\|_{W_1} \right)$ are separable.

*Proof.* Recall that $(\mathscr{P}, W_1)$ is separable [Theorem 6.18 Villani et al., 2009], and by Lemma E.1, the Cramér distance $\ell_2$ can be bounded by 1-Wasserstein distance $W_1$, hence $(\mathscr{P}, \ell_2)$ is also separable. Let $d$ be either $W_1$ or $\ell_2$, and $A$ be the countable dense subset of $(\mathscr{P}, d)$. For any $\epsilon > 0, \mu \in \mathcal{M}$, we denote $\tilde{\mu} := \frac{2\mu}{|\mu|(\mathbb{R})}$. Then we can find a $q \in \mathbb{Q}, \bar{\mu}_+, \bar{\mu}_- \in A$, s.t. $\left| q - \frac{1}{2} |\mu|(\mathbb{R}) \right| \leq \frac{\epsilon |\mu|(\mathbb{R})}{6 \|\mu\|_d}$, $d(\bar{\mu}_+, \tilde{\mu}_+) \leq \frac{\epsilon}{3q}, d(\bar{\mu}_-, \tilde{\mu}_-) \leq \frac{\epsilon}{3q}$. Note that the set of all possible $\bar{\mu} = \bar{\mu}_+ - \bar{\mu}_-$ is countable. Let $\hat{\mu} := q\bar{\mu}$, then we have

$$\|\mu - \hat{\mu}\|_d$$

$$\leq \|\mu - q\tilde{\mu}\|_d + \|q\tilde{\mu} - \hat{\mu}\|_d$$

$$= \left\| (\frac{1}{2} |\mu|(\mathbb{R}) - q)\tilde{\mu} \right\|_d + q \|(\tilde{\mu}_+ - \bar{\mu}_+) - (\tilde{\mu}_- - \bar{\mu}_-)\|_d$$

$$\leq \left| q - \frac{1}{2} |\mu|(\mathbb{R}) \right| \frac{2 \|\mu\|_d}{|\mu|(\mathbb{R})} + q \|\tilde{\mu}_+ - \bar{\mu}_+\|_d + q \|\tilde{\mu}_- - \bar{\mu}_-\|_d$$

$$\leq \frac{\epsilon |\mu|(\mathbb{R})}{6 \|\mu\|_d} \frac{2 \|\mu\|_d}{|\mu|(\mathbb{R})} + q \frac{\epsilon}{3q} + q \frac{\epsilon}{3q}$$

$$= \epsilon.$$

Hence we have found a countable dense subset for $\left( \mathcal{M}, \|\cdot\|_d \right)$ for $d = W_1$ or $\ell_2$. Therefore, $\left( \mathcal{M}, \|\cdot\|_{\ell_2} \right)$ and $\left( \mathcal{M}, \|\cdot\|_{W_1} \right)$ are separable. $\square$

**Lemma E.3.** $(\mathcal{M}, \|\cdot\|_{W_1})$ *is not complete.*

*Proof.* Consider the Cauchy sequence in $(\mathcal{M}, \|\cdot\|_{W_1})$ $\mu_n = \sum_{i=1}^{n} \left( \delta_{\frac{1}{i} + \frac{1}{2^i}} - \delta_{\frac{1}{i} - \frac{1}{2^i}} \right)$, which satisfies $\|\mu_n - \mu_{n+k}\|_{W_1} \leq \sum_{i=1}^{k} \frac{1}{2^{n+i-1}} \leq \frac{1}{2^{n-1}} \to 0$, but its limit $\sum_{i=1}^{\infty} \left( \delta_{\frac{1}{i} + \frac{1}{2^i}} - \delta_{\frac{1}{i} - \frac{1}{2^i}} \right) \notin \mathcal{M}$ because its total variation is infinity. Hence, this is a Cauchy sequence without a limit, implying the space is not complete. $\qquad\square$

**Lemma E.4** (Range of $\eta_t^\pi$). *Suppose that $\alpha_t \in [0, 1]$ for all $t \geq 0$. Assume that $\eta_0^\pi \in \mathscr{P}^\mathcal{S}$, then we have, for all $t \geq 0$, $\eta_t^\pi \in \mathscr{P}^\mathcal{S}$ in the case of NTD. Similarly, assume that $\eta_0^\pi \in \mathscr{P}_K^\mathcal{S}$, then we have, for all $t \geq 0$, $\eta_t^\pi \in \mathscr{P}_K^\mathcal{S}$ in the case of CTD.*

*Proof.* We will only prove the case of NTD, and the proof for CTD is similar by utilizing the property of the projection operator $\Pi_K \colon \mathscr{P}^\mathcal{S} \to \mathscr{P}_K^\mathcal{S}$. We prove the result by induction. It is trivial that $\eta_t^\pi \in \mathscr{P}^\mathcal{S}$ for $t = 0$. Suppose that $\eta_{t-1}^\pi \in \mathscr{P}^\mathcal{S}$, recall the updating scheme of NTD

$$\eta_t^\pi = (1 - \alpha_t)\eta_{t-1}^\pi + \alpha_t \mathcal{T}_t^\pi \eta_{t-1}^\pi. \tag{78}$$

It is evident that $\mathscr{P}^\mathcal{S}$ is a convex set, considering that $\mathscr{P}^\mathcal{S}$ is a subset of the product signed measure space, which is a linear space. Therefore, we only need to show that $\mathcal{T}_t^\pi \eta_{t-1}^\pi \in \mathscr{P}^\mathcal{S}$, which trivially holds since $\mathcal{T}_t^\pi$ is a random operator mapping from $\mathscr{P}^\mathcal{S}$ to $\mathscr{P}^\mathcal{S}$, and $\eta_{t-1}^\pi \in \mathscr{P}^\mathcal{S}$. By applying the induction argument, we can arrive at the conclusion. $\qquad\square$

