# OpenReview forum: "Statistical Efficiency of Distributional Temporal Difference Learning"
_NeurIPS.cc/2024/Conference — NeurIPS 2024 oral_

### Official Review · Reviewer_KRno · 2024-07-04

**Soundness:** 3
**Presentation:** 2
**Contribution:** 1
**Rating:** 3
**Confidence:** 4

**Summary:**

This paper studies the finite sample performance/non-asymptotic analysis of distributional temporal difference by providing a tighter (minimax optimal) bound than previous works. They propose the non-parametric distributional TD without incorporating any parameterization error. By leveraging the conclusion that the zero-mass signed measure space with the cramer metric is a Hilbert space, the authors propose a novel Freedman’s inequality (Theorem A.2) in the stochastic approximation literature, which is used to derive the sample complexity results for the distributional TD.

**Strengths:**

* It is reasonable to investigate the non-asymptotic convergence of distributional TD.

* The resulting conclusions in Theorems 4.1 and 4.2 are technically sound based on the stochastic approximation proof technique.

**Weaknesses:**

* The theoretical contribution is within a limited scope. Since the asymptotic and non-asymptotic convergence of distributional TD have already been investigated, this paper only contributes to providing tighter non-asymptotic bounds (by using a generative model), which is thus limited in the research scope. The current version is not well-motivated to me.

* The proposed non-parametric distributional TD is impractical, which is mainly helpful for theoretical results. As CTD also follows the same updating rule, the theoretical results in Sections 4.1 and 4.2 apply in a parallel manner, which may not be novel.

* The proposed Freedman’s inequality is one kind of invariant of its vanilla version, which may be sufficiently valued, as far as I can tell. If the authors think this inequality should be viewed as a main theoretical contribution, it should be emphasized and submitted as a stochastic approximation paper in applied probability instead of RL. As for the current version, I acknowledge that this inequality is useful for the finite sample analysis, but the authors are suggested to put more thought into how to posit this contribution more reasonably.

* The writing needs substantial improvement. While it is basically clear, I find it hard to understand some parts of the paper. For example, CTD uses the approximation in (8), where $\eta$ should be $\hat{\eta}$. Some related works are missing beyond QTD and CTD, such as kernel-based or sample-based TD methods [1].

[1] Distributional Bellman Operators over Mean Embeddings (ICML 2024)

**Questions:**

How to show the bounds in Theorems 4.1 and 4.2 are minimax optimal? Is it possible to conduct some toy examples to verify them?

**Limitations:**

Some assumptions may not be clearly stated in the main content. It is not very clear in what detailed conditions the derived bounds are minimax optimal.

---

> ### Author Rebuttal · Authors · 2024-08-07
>
> We sincerely appreciate the reviewer's time and effort in reviewing our paper.
> We are glad to hear that the reviewer finds it reasonable to investigate the non-asymptotic convergence of distributional TD.
> We are also happy to know the reviewer thinks our theoretical results are technically sound.
> Below, we hope our rebuttal can address the reviewer's concerns.
>
> > Weakness 1: Since the asymptotic ... limited in the research scope.
>
> We respectfully disagree with the reviewer on this point. From our perspective,
> many important works in the ML theory community focus on improving existing
> theoretical results. We would like to stress that our improvement is especially
> significant as we actually prove the near-minimax-optimal sample complexity
> bounds. We would also like to clarify the problem setting in our paper. The
> setting we adopt is called the synchronous setting, which can be widely seen in
> the RL theory literature [Wainwright, 2019, Li et al., 2024]. Such a setting is
> also adopted in the asymptotic and non-asymptotic analysis of distributional TD
> as mentioned by the reviewer [Rowland et al., 2018, 2023, Bock and Heitzinger,
> 2022]. See also the sample complexity table of DRL in https://openreview.net/forum?id=eWUM5hRYgH&noteId=5fyooKmeuH.
>
> > Weakness 2: The proposed ... may not be novel.
>
> As we claim in the paper, NTD is only a simplified, conceptual algorithm and we use it to make our theoretical analysis more accessible to readers.
> In contrast, the CTD algorithm is widely used in practical applications, and our non-asymptotic bounds help to build a solid theoretical understanding of its performance.
> We feel a bit confused as the reviewer states that NTD is not significant because it is impractical and the analysis of CTD is not significant because it is parallel to the analysis of NTD.
> We kindly request the reviewer to add further explanations as we feel we may misunderstand the reviewer's concerns.
>
> > Weakness 3: The proposed Freedman’s ... contribution more reasonably.
>
> We are glad to hear the reviewer acknowledged that the proposed Freedman’s
> inequality can be a contribution of independent interest. But we kindly disagree
> with the reviewer’s comment that the proposed Freedman’s inequality can be
> viewed as a main theoretical contribution only in a stochastic approximation
> paper in applied probability instead of RL. As the novel Freedman’s inequality
> is developed as a key technique tool for our theoretical analysis, we feel
> it can be an important theoretical contribution of this paper. And we thank
> the reviewer for the suggestion that we should better posit such a contribution
> (possibly following the advice of Reviewer bMoZ to add it to the main paper).
>
> > Weakness 4: The writing ... sample-based TD methods.
>
> We thank the reviewer for the comment about the typo and have fixed it.
> We would appreciate it if the reviewer could point out the parts he/she feels are
> hard to understand, which could greatly help us to improve the quality of our
> paper. We are glad to include the paper [Wenliang et al., 2023] in our literature
> review. We would also discuss more papers on sample-based TD as the reviewer
> suggests.
>
> > Questions: How to show ... verify them?
>
> Since the value function $V^\pi(s)=\mathbb{E}_{G\sim\eta^\pi(s)}\left[G\right]$, and $W_1$ metric satisfies
>
> $W_1(\mu,\nu)=\sup_{f\colon Lip(f) \leq 1} \|\mathbb{E}[f(X)] -\mathbb{E}[f(Y)] \|$, where $X\sim\mu$ and $Y\sim\nu$,
>
> we always have
>
> $ \sup_{s}\|\widehat V^\pi(s)-V^\pi(s)\|\leq \sup_{s}W_1(\eta^\pi(s),\hat\eta^\pi(s))    $
>
> Therefore, any lower bound for the problem of standard policy evaluation would be a valid lower bound for the problem we consider.
> Since $\widetilde{\Omega}\left(\frac{1}{\varepsilon^2 (1-\gamma)^3}\right)$ is a lower bound for the standard policy evaluation (see [Pananjady and Wainwright,
> 2020], Theorem 2(b)), it is also a lower bound for our problem.
> Therefore, the $\widetilde{O}\left(\frac{1}{\varepsilon^2 (1-\gamma)^3}\right)$ upper bound we show matches the lower bound (up to logarithmic factors) and is thus near-minimax-optimal.
>
> We would like to express our gratitude for your inquiry. We will revise the manuscript to clarify this point and ensure that our exposition is as precise and understandable as possible.
>
> > Limitations: Some assumptions ... minimax optimal.
>
> We believe that we have clearly stated the theoretical assumptions in the main text, and the points made by other reviewers: "This paper is very technically precise" (bMoZ) and "The paper clarifies the limitations by making assumptions for theoretical explanations" (CFYY) also support this view.
> We would be grateful if the reviewer could specify which assumptions have not been explicitly stated, as this would greatly assist us in enhancing the quality of our work.
> The topic of minimax-optimality has already been discussed in our response to the questions part.
>
> ## References
> Mohammad Gheshlaghi Azar, R´emi Munos, and Hilbert J Kappen. Minimax pac
> bounds on the sample complexity of reinforcement learning with a generative
> model. Machine learning, 91:325–349, 2013.
>
> Ashwin Pananjady and Martin J Wainwright. Instance-dependent ℓ∞-bounds
> for policy evaluation in tabular reinforcement learning. IEEE Transactions on
> Information Theory, 67(1):566–585, 2020.
>
> Martin J Wainwright. Stochastic approximation with cone-contractive operators:
> Sharp ℓ∞-bounds for q-learning. arXiv preprint arXiv:1905.06265, 2019.

---

> ### Comment · Area_Chair_aGjp · 2024-08-11
> **Please respond to the authors**
>
> Hello reviewer KRno: The authors have responded to your comments. I would expect you to respond in kind.

---

### Official Review · Reviewer_CFYY · 2024-07-11

**Soundness:** 3
**Presentation:** 3
**Contribution:** 3
**Rating:** 7
**Confidence:** 3

**Summary:**

The paper investigates the statistical efficiency of distributional temporal difference (TD) algorithms in reinforcement learning, focusing on non-asymptotic results. The authors introduce a non-parametric distributional TD algorithm (NTD), analyze its sample complexity with respect to the p-Wasserstein metric and Cramer metric, and show the near minimax optimality. They also revisit the categorical TD (CTD) and prove that it shares similar non-asymptotic convergence bounds with NTD.

**Strengths:**

- The notation is standard and clear, and the related work on sample complexity of distributional RL is well-written, making it easy to understand the authors' theoretical contributions.

- The proposed analysis of NTD and CTD provides a tighter near-minimax optimal sample complexity bound compared to previous methods.

- A new Freedman's inequality is presented, which seems to be useful for further research.

- The bound on $\beta_k^{(t)}$ presented in Eq. (18) is impressive and appears highly useful.

**Weaknesses:**

As a minor comment, it would be helpful to present sample complexities of this paper and related work in a table to help understand the authors' contributions.

The matrix-wise distributional bellman operator $\mathcal{T}(s,s')$ presented on line 257-259 uses the same notation as the standard, which can be misleading.

Typos:
- In Line 105, the font for $T^{\pi}$ should be corrected to calligraphic.
- In lines 211,212, $K \geq \frac{4}{\epsilon^2 (1-\gamma)^3}$ should be $K \geq \frac{4}{\epsilon^2 (1-\gamma)^2}$.
- On line 275, the period is written twice.
- Line 510 should be corrected to $\log$, not $\log_2$.

**Questions:**

- In Theorem 4.2, I am curious about the derivation of the statement that the number of atoms required for CTD to converge is $\tilde{O}(1/\epsilon^2 (1-\gamma)^2)$.

- Additionally, DCFP also achieves the same sample complexity with categorical representation, so a detailed explanation of the differences between these two papers would be beneficial.

**Limitations:**

The paper clarifies the limitations by making assumptions for theoretical explanations.

---

> ### Author Rebuttal · Authors · 2024-08-07
>
> Thank you for your encouraging comments.
> We are very glad to know that
> you find our results solid and novel.
> Regarding the weaknesses and
> questions, we provide the following detailed responses:
>
> > Weakness 1: As a minor ... understand the authors' contributions.
> Thanks for your suggestion of presenting a sample complexities table, we agree with
> you that it will help readers better understand the results in our paper.
>
> We will add the following table in the revised version.
> In the table, when the task is distributional policy evaluation, the sample complexity is defined in terms of the $W_1$ metric as the measure of error.
> This allows for a clearer comparison of results with those in the standard policy evaluation task.
> | Paper | Sample Complexity | Method | Task|
> |-------|-------|-------|-------|
> |[Gheshlaghi Azar et al., 2013] | $\widetilde{O}\left(\frac{1}{\varepsilon^2 (1-\gamma)^3}\right)$ | Model-based | Policy evaluation|
> |[Li et al., 2024] | $\widetilde{O}\left(\frac{1}{\varepsilon^2 (1-\gamma)^3}\right)$ | TD (Model-free) | Policy evaluation|
> |[Rowland et al., 2024] | $\widetilde{O}\left(\frac{1}{\varepsilon^2 (1-\gamma)^3}\right)$ | DCFP (Model-based) | Distributional policy evaluation|
> |[Rowland et al., 2018] | Asymptotic | CTD (Model-free) | Distributional policy evaluation|
> |[Rowland et al., 2023] | Asymptotic | QTD (Model-free) | Distributional policy evaluation|
> |[Bock and Heitzinger, 2022] | $\widetilde{O}\left(\frac{1}{\varepsilon^2 (1-\gamma)^4}\right)$ | SCPE (Model-free) | Distributional policy evaluation|
> |this work | $\widetilde{O}\left(\frac{1}{\varepsilon^2 (1-\gamma)^3}\right)$ | CTD (Model-free) | Distributional policy evaluation|
>
> Here, DCFP method proposed by [Rowland et al., 2024] can be seen as an extension of the certainty
> equivalence method (also called model-based approach) in traditional RL to the
> domain of DRL. And SCPE proposed by [Bock and Heitzinger, 2022] can be
> regarded as CTD with an additional acceleration term.
>
> > Weakness 2: The matrix-wise ... can be misleading
>
> We appreciate your valuable suggestions.
> In the subsequent versions, we will consider revising the notation, such as $\mathcal{T}_{s,s^\prime}$, to prevent any potential misunderstanding by the readers.
>
> > Typos
>
> We are grateful for your careful reading and for identifying these typos.
> We will correct them in the subsequent version of the manuscript.
>
> However, regarding the third point you raised, on lines 211 and 212, where it states $K\geq\frac{4}{\varepsilon^2(1-\gamma)^3}$
> , there is no typographical error.
> This is because we require $\bar{W}_1(\eta^{\pi,K},\eta^{\pi})\leq\frac{\varepsilon}{2}$ to hold, and since $\bar{W}_1(\eta^{\pi,K},\eta^{\pi})\leq\frac{1}{\sqrt{1-\gamma}}\bar{\ell}_2(\eta^{\pi,K},\eta^{\pi})\leq\frac{1}{\sqrt{K}(1-\gamma)^{3/2}}$ according to Equation (7), we need to take $K\geq\frac{4}{\varepsilon^2(1-\gamma)^3}$.
> And we would like to clarify a related typo in line 206 (Theorem 4.2), 297, 577: $K\geq\frac{4}{\varepsilon^2(1-\gamma)^2}$ should be corrected to $K\geq\frac{4}{1-\gamma}$.
> We sincerely apologize for the confusion this may have caused.
>
> > Question 1: In Theorem 4.2, I am curious about ... $\widetilde{O}\left(\frac{4}{\varepsilon^2(1-\gamma)^2}\right)$
>
> As previously mentioned, in the conditions for Theorem 4.2, the statement $K\geq\frac{4}{\varepsilon^2(1-\gamma)^2}$ should be corrected to $K\geq\frac{4}{1-\gamma}$.
> We sincerely apologize for this mistake again.
> In the proof of Theorem 4.2, the condition $K\geq\frac{4}{1-\gamma}$ is only utilized in the proof of Lemma B.3 (see Equation (81)), which ensures that the variance term $\|\|(I-\gamma P)^{-1}\sigma(\eta)\|\|$ can be finely controlled by $\frac{2}{1-\gamma}$, while the naive upper bound is $\frac{1}{(1-\gamma)^{2}}$.
> This is because when $K>\frac{4}{1-\gamma}$, the variance term under the CTD setting approaches that under the NTD setting, allowing us to derive the tight sample complexity bound.
> Specifically, following the proof of Corollary 5.12 in page 20 of [Rowland et al., 2024], if we take $K>\frac{4}{1-\gamma}$, we also have $\frac{2}{K\sqrt{1-\gamma}}+\frac{1}{K^2(1-\gamma)^2}<\frac{1}{2}\sqrt{1-\gamma}+\frac{1}{16}<1$, which leads to the desired conclusion $\|\|(I-\gamma P)^{-1}\sigma(\eta)\|\|\leq\frac{2}{1-\gamma}$.
>
> > Question 2: Additionally, DCFP ... be beneficial.
>
> We appreciate your valuable suggestion, and we will provide a
> further comparison with [Rowland et al., 2024]. The DCFP method proposed by
> [Rowland et al., 2024] can be seen as an extension of the certainty equivalence
> method (also called model-based approach) in traditional RL to the domain of
> DRL. In DCFP, one needs to estimate the distributional Bellman operator using
> all samples, and then substitute it for the ground-truth one in the distributional
> Bellman equation to solve for the estimator of $\eta^\pi$This can be considered as a
> plug-in estimator, which is less similar to practical algorithms.
>
> In contrast, the CTD analyzed in this paper can be viewed as an extension
> of TD. Compared to DCFP, CTD is more similar to practical algorithms and
> involves a more complex analysis.
>
> In terms of proof techniques, [Rowland et al., 2024] introduced the important
> tools: the stochastic categorical CDF Bellman equation, to derive tight sample
> complexity bounds for the model-based method DCFP. The tools are also used
> in our paper. Compared to DCFP, the analysis of CTD (a model-free method)
> is more challenging. For instance, some probabilistic tools used for analyzing
> stochastic approximation problems in Hilbert spaces are not available, such as
> the Freedman’s inequality. We overcame these difficulties and obtained tight
> sample complexity bounds. We believe that our findings will be of interest to
> researchers working on distributional reinforcement learning and related areas.
>
> We will revise the manuscript to include the discussion above, which we
> believe will provide a clearer context for our work.

---

> > ### Comment · Reviewer_CFYY · 2024-08-12
> >
> > Thank you for the detailed response.
> > However, I am still not convinced by the authors' answer to Question 1.
> > According to the revised conclusion, it seems that it is possible for the number of atoms $K = 4/(1-\gamma) + 1$ to converge to a unique fixed point, $\eta^\pi$, for sufficiently large $T$, independent of the $\epsilon$.
> > However, $\eta^{\pi, K}$ is a distribution represented by a finite number of $K$ representations, leading to the incorrect conclusion that the discrepancy with $\eta^\pi$ can be reduced by $\epsilon$ with a fixed $K$.
> > Could the author provide further clarification on this issue?

---

> > > ### Author Response · Authors · 2024-08-12
> > >
> > > Thank you very much for your response and new question. We sincerely apologize that our statements in Section 4.2 and replies were unclear and misleading.
> > > We would like to clarify this point in more details.
> > >
> > > In Theorem 4.2 (line 209), our conclusion is that when $K>\frac{4}{1-\gamma}$, we can choose step size $\alpha_t$ and total update steps $T$ independent of $K$ such that the $W_1$ distance between the estimator $\eta^\pi_T$ and
> > > $\eta^{\pi,K}$ _(instead of $\eta^\pi$)_ $\bar{W}_1(\eta^\pi_T,\eta^{\pi,K})\leq \frac{\varepsilon}{2}$ (both are distributions represented by a finite number of $K$ atoms), we call this estimation error.
> > > In line 210-212, we deal with the approximation error $\bar{W}_1(\eta^{\pi,K},\eta^\pi)$, which is less than $\frac{\epsilon}{2}$ when $K\geq\frac{4}{\varepsilon^2(1-\gamma)^3}$ (now the condition $K> \frac{4}{1-\gamma}$ in Theorem 4.2 naturally holds) according to Equation (7).
> > > In summary, when $K\geq\frac{4}{\varepsilon^2(1-\gamma)^3}$, we have the desired conclusion $\bar{W}_1(\eta^\pi_T,\eta^\pi)\leq\bar{W}_1(\eta^\pi_T,\eta^{\pi,K})+\bar{W}_1(\eta^{\pi,K},\eta^\pi)\leq \epsilon$.
> > >
> > > In short, Theorem 4.2 only deals with the estimation error $\bar{W}_1(\eta^\pi_T,\eta^{\pi,K})$, and line 210-212 deal with the approximation error $\bar{W}_1(\eta^{\pi,K},\eta^\pi)$.
> > >
> > > We will revise the manuscript to clarify these points and add a more detailed explanation to ensure the argument is clear and logical for the readers.
> > > We would like to express our gratitude once again for the reviewer's insightful question.

---

> > > > ### Comment · Reviewer_CFYY · 2024-08-12
> > > >
> > > > Thank you for the author's quick response.
> > > > The theoretical results that I thought were ambiguous have been clarified sufficiently, and I hope this is reflected appropriately in the paper. Therefore, I will raise the score to 7.

---

> > > > > ### Author Response · Authors · 2024-08-12
> > > > > **Thank you!**
> > > > >
> > > > > We are deeply appreciative of your insightful and beneficial comments on our manuscript. We will incorporate your suggestions (especially modify the statements in line 205-213 including Theorem 4.2) in the revised version.

---

> ### Comment · Area_Chair_aGjp · 2024-08-11
> **Please respond to the authors**
>
> Hello reviewer CFYY: The authors have responded to your comments. I would expect you to respond in kind.

---

### Official Review · Reviewer_bMoZ · 2024-07-12

**Soundness:** 3
**Presentation:** 2
**Contribution:** 3
**Rating:** 7
**Confidence:** 3

**Summary:**

This paper presents last-iterate error bounds for distributional
temporal difference learning in the $W_p$ and Cramér metrics. The
results apply to a nonparametric/intractable distributional TD algorithm
(where return distributions can be represented exactly) and a tractable
projected distributional TD algorithm with finite categorical
parameterizations of return distributions. Using a novel Hilbert space
martingale inequality, the paper achieves tighter bounds than existing
results in the literature, as well as generalizations to more metric
spaces.

**Strengths:**

This paper is very technically precise, and mostly well written. The Hilbert
space Freedman inequality that was derived for the purpose of proving several
results in the paper seems useful. Moreover, the error bounds generalize and/or
improve upon the existing results on non-asymptotic sample complexity in
distributional RL.

**Weaknesses:**

Many of the mathematical derivations were very quick and at times
difficult to follow (particularly, in the appendix). While the work of
Rowland et al. 2024 studies an algorithm that is less similar to
practical distributional RL methods, the conclusion of their work is
fairly similar (distributional RL has the same statistical efficiency as
expected-value RL). Their work does not cover all $W_p$ distances unlike
this work, and the dependence on $p$ is interesting; it is also
interesting, again, that this work provides a certificate for good
statistical efficiency with a stochastic approximation TD algorithm.
That said, the paper does not do an excellent job of motivating why this
novelty is exciting.

Moreover, since the Hilbert space Freedman inequality appears to be an
important contribution, I believe this should have been included in the
main text (even just the statement of the theorem).

Finally, I suspect there is a slight mathematical mistake in Appendix A,
though I don't expect it substantially changes the results; see the
Questions section.

## Minor Issues

"Temporal Difference" should really be "Temporal Difference Learning".

Line 159 says "the projection is unique and uniquely given by" – this is
a little redundant, should be either "the projection is uniquely given
by" or "the projection is unique and is given by".

Line 168 mentions the BLT theorem without any citation — it might be
nice to include a reference here, since "BLT theorem" may not be a
familiar term to some.

At the end of line 275, there are two periods.

On line 425, "forth" -\> "fourth".

**Questions:**

In Theorem 4.2, you claim that the sample complexity bound does not
depend on the number of bins $K$. However, you also assume an explicit
lower bound on $K$, so naturally the sample complexity bound does depend
on $K$ in some capacity. Can anything be said about the sample
complexity (e.g., as a function of $K$) when
$K\leq 4\epsilon^{-2}(1-\gamma)^{-2}$?

Is the derivative computed on the second step of equation (29) correct?
It looks like you went from $\lambda$ to $\lambda^2$, but I believe only
on of the terms on the RHS should incur an extra $\lambda$ factor. My
computation is

\begin{align*}
\phi''(t) &= \lambda\mathbb{E}\_{j-1}\left\\{\frac{\mathrm{d}}{\mathrm{d}t}[\sin(\lambda u(t))u'(t)]\right\\}\\\\
&= \lambda\mathbb{E}_{j-1}\left(u'(t)\frac{\mathrm{d}}{\mathrm{d} t}\sinh(\lambda u(t)) + \sinh(\lambda u(t))\frac{\rm d}{\mathrm{d}t}u'(t)\right)\\\\
&= \lambda\mathbb{E}\_{j-1}\left(\lambda (u'(t))^2\cosh(\lambda u(t)) + \sinh(\lambda u(t))u''(t)\right).
\end{align*}

**Limitations:**

Limitations are adequately discussed.

---

> ### Author Rebuttal · Authors · 2024-08-07
>
> Thank you for your valuable review and constructive suggestions.
> We are very glad to know thatyou think our theoretical results are technically precise, and acknowledge the contribution of the proposed Freedman's inequality.
>
> Regarding the weaknesses and questions, we provide the following detailed responses:
> > Weakness 1: Many of the mathematical ... (particularly, in the appendix).
>
> We thank the reviewer's comment regarding the presentation of the mathematical derivations. We will add more explanations to make our proof easier to follow.
>
> > Weakness 2: While the work of Rowland et al. 2024 ... why this novelty is exciting.
>
> We are grateful to the reviewer for raising this issue, and we will provide a
> further comparison with [Rowland et al., 2024]. The DCFP method proposed by
> [Rowland et al., 2024] can be seen as an extension of the certainty equivalence
> method (also called model-based approach) in traditional RL to the domain of
> DRL. In DCFP, one needs to estimate the distributional Bellman operator using
> all samples, and then substitute it for the ground-truth one in the distributional
> Bellman equation to solve for the estimator of $\eta^\pi$This can be considered as a
> plug-in estimator, which is less similar to practical algorithms.
>
> In contrast, the CTD analyzed in this paper can be viewed as an extension
> of TD. Compared to DCFP, CTD is more similar to practical algorithms and
> involves a more complex analysis.
>
> In terms of proof techniques, [Rowland et al., 2024] introduced the important
> tools: the stochastic categorical CDF Bellman equation, to derive tight sample
> complexity bounds for the model-based method DCFP. The tools are also used
> in our paper. Compared to DCFP, the analysis of CTD (a model-free method)
> is more challenging. For instance, some probabilistic tools used for analyzing
> stochastic approximation problems in Hilbert spaces are not available, such as
> the Freedman’s inequality. We overcame these difficulties and obtained tight
> sample complexity bounds. We believe that our findings will be of interest to
> researchers working on distributional reinforcement learning and related areas.
>
> We will revise the manuscript to include the discussion above, which we
> believe will provide a clearer context for our work.
>
> > Weakness 3: Moreover, since the Hilbert space ... just the statement of the theorem).
>
> We thank the reviewer for recognizing the contribution of the Hilbert space
> Freedman’s inequality. We will add a new section after the Background to
> include the inequality in the revised manuscript.
>
> > Weakness 4: Finally, I suspect there is a slight mathematical mistake...
>
> We thank the reviewer for spotting the typographical error. Equation (29)
> should be corrected to
>
> $ \phi^{\prime\prime}(t)=\lambda\mathbb{E}_{j-1}\left\lbrace\frac{d}{dt}\left[\sinh\left(\lambda u(t)\right)u^\prime(t)\right]\right\rbrace$
>
> $ =\lambda \mathbb{E}_{j-1}\left[\lambda\left(u^\prime(t)\right)^2\cosh\left(\lambda u(t)\right)+u^{\prime\prime}(t)\sinh\left(\lambda u(t)\right)\right] $
>
> $ \leq \lambda^2 \mathbb{E}_{j-1}\left[\left(\left(u^\prime(t)\right)^2+u^{\prime\prime}(t)u(t)\right)\cosh\left(\lambda u(t)\right)\right] $
>
> $ =\cdots$
>
> where in the third line, we used $\sinh(\lambda u(t))\leq \lambda u(t)\cosh(\lambda u(t))$ (since $\lambda>0$ and $h(x)=x\cosh(x)-\sinh(x)\geq 0$ for any $x\geq 0$), and $u^{\prime\prime}(t)=\frac{\|\|X_j\|\|^2u(t)-\frac{\langle Y_{j-1}+tX_j,X_j\rangle^2}{u(t)}}{u^2(t)}\geq 0$ by Cauchy-Schwarz inequality.
> No further modifications are needed for subsequent proofs.
>
> > Minor Issues
>
> We are grateful to the reviewer for identifying these issues in our manuscript.
> We have fixed them and added a reference to the BLT theorem ([Hunter and
> Nachtergaele, 2001], Theorem 5.19).
>
> > Question: In Theorem 4.2, you claim that the sample complexity bound does not depend on the number of bins $K\cdots$
>
> We appreciate the reviewer's questions and are sorry that our original statement was unclear.
> The original idea that once
> $K> \frac{4}{1-\gamma}$ ($K>\frac{4}{\varepsilon^2(1-\gamma)^2}$ in line 206, 297, 577 is a typo), we can choose step size $\alpha_t$ and total update steps $T$ independent of $K$ to get the desired conclusion.
> When proving Theorem 4.2, the condition $K> \frac{4}{1-\gamma}$ is only utilized in the proof of Lemma B.3 (see Equation (81)), which ensures that the variance term $\|\|(I-\gamma P)^{-1}\sigma(\eta)\|\|$ can be finely controlled by $\frac{2}{1-\gamma}$, while the naive upper bound is $\frac{1}{(1-\gamma)^{2}}$.
> This is because when $K> \frac{4}{1-\gamma}$, the variance term under the CTD setting approaches that under the NTD setting, allowing us to derive that $T=\widetilde{O}\left(\frac{1}{\varepsilon^2 (1-\gamma)^3}\right)$ is sufficient to make sure $\bar{W}_1\(\eta^\pi_T,\eta^{\pi,K})\leq \varepsilon$.
> When $K\leq \frac{4}{1-\gamma}$, we can obtain a sub-optimal result: $T=\widetilde{O}\left(\frac{1}{\varepsilon^2 (1-\gamma)^4}\right)$ is sufficient to make sure $\bar{W}_1(\eta^\pi_T,\eta^{\pi,K})\leq \varepsilon$, through an analysis that does not use variance information (e.g. Hilbert space Azuma-Hoeffding inequality).
>
> In fact, Theorem 4.2 only guarantees $\bar{W}_1(\eta^\pi_T,\eta^{\pi,K})\leq \frac{\varepsilon}{2}$.
> To ensure that the desired error term $\bar{W}_1(\eta^\pi_T,\eta^{\pi})\leq\varepsilon$, we need $\bar{W}_1(\eta^{\pi,K},\eta^{\pi})\leq\frac{1}{\sqrt{1-\gamma}}\bar{\ell}_2(\eta^{\pi,K},\eta^{\pi})\leq\frac{\varepsilon}{2}$, at which point $K\geq\frac{4}{\varepsilon^2(1-\gamma)^3}$, and therefore the condition $K> \frac{4}{1-\gamma}$ in Theorem 4.2 naturally holds.
>
> We will revise the manuscript to clarify these points and add a more detailed
> explanation to ensure the argument is clear and logical for the readers. We would
> like to express our gratitude once again for the reviewer’s insightful question.
>
> ## Reference
> John K Hunter and Bruno Nachtergaele. Applied analysis. World Scientific
> Publishing Company, 2001.

---

> > ### Comment · Reviewer_bMoZ · 2024-08-08
> >
> > Thanks to the reviewers for their detailed response. I think this paper makes a really nice contribution to the analysis of distributional TD (especially once the proofs are made easier to follow), and I will raise my score to reflect this.

---

> > > ### Author Response · Authors · 2024-08-09
> > > **Thank you!**
> > >
> > > We would like to express our sincere gratitude again for your constructive and valuable comments suggestions on our paper. We will incorporate the suggestions (especially add more explanations to make our proof easier to follow) in the revised version.

---

### Decision · Program_Chairs · 2024-09-25

**Decision:**

Accept (oral)

**Comment:**

The reviewers are generally strongly positive that this paper makes substantive contributions to understanding the sample complexity of distributional RL that are enabled by interesting technical contributions. I believe this paper will be of broad interest to the community.